

# Rainfall–Runoff Prediction at Multiple Timescales with a Single Long Short-Term Memory Network

Martin Gauch[1, 2], Frederik Kratzert[1], Daniel Klotz[1], Grey Nearing[3], Jimmy Lin[2], and Sepp Hochreiter[1]

[1]Institute for Machine Learning, Johannes Kepler University Linz, Linz, Austria
[2]David R. Cheriton School of Computer Science, University of Waterloo, Waterloo, Canada
[3]Google Research, Mountain View, CA, USA

**Correspondence:** Martin Gauch (gauch@ml.jku.at)

**Abstract.** Long Short-Term Memory Networks (LSTMs) have been applied to daily discharge prediction with remarkable success. Many practical scenarios, however, require predictions at more granular timescales. For instance, accurate prediction of short but extreme flood peaks can make a life-saving difference, yet such peaks may escape the coarse temporal resolution of daily predictions. Naively training an LSTM on hourly data, however, entails very long input sequences that make learning

hard and computationally expensive. In this study, we propose two Multi-Timescale LSTM (MTS-LSTM) architectures that jointly predict multiple timescales within one model, as they process long-past inputs at a single temporal resolution and branch out into each individual timescale for more recent input steps. We test these models on 516 basins across the continental United States and benchmark against the US National Water Model. Compared to naive prediction with a distinct LSTM per timescale, the multi-timescale architectures are computationally more efficient with no loss in accuracy. Beyond prediction quality, the

multi-timescale LSTM can process different input variables at different timescales, which is especially relevant to operational applications where the lead time of meteorological forcings depends on their temporal resolution.

## 1   Introduction

Rainfall–runoff modeling approaches based on deep learning—particularly Long Short-Term Memory (LSTM) networks—have proven highly successful in a number of studies. LSTMs can predict multiple catchments using a single model and yield

more accurate predictions than state-of-the-art process-based models in a variety of benchmarks (Kratzert et al., 2019).

Different applications require hydrologic information at different timescales. For example, hydropower operators might care about daily or weekly (or longer) inputs into their reservoirs, while flood forecasting requires sub-daily predictions. Yet, much of the work in applying deep learning to streamflow prediction has been at the daily timescale. Daily forecasts make sense for medium- to long-range forecasts, however, daily input resolution mutes diurnal variations that may cause important variations

in discharge signatures, such as evapotranspiration and snowmelt. Thus, daily predictions are often too coarse to provide actionable information for short-range forecasts. For example, in the event of flooding, the distinction between moderate discharge spread across the day and the same amount of water compressed into a few hours of flash flooding may pose a life-threatening difference.



Because of this, operational hydrologic models often operate at multiple timescales using several independent setups of
a traditional, process-based rainfall–runoff model. For instance, the US National Oceanic and Atmospheric Administration's
(NOAA) National Water Model (NWM) produces hourly short-range forecasts every hour, as well as three- and six-hourly
medium- to long-range forecasts every six hours.[1] This approach multiplies the computational resource demand and can result
in inconsistent predictions wherever two setups overlap in their predicted time ranges.

The issue of multiple input and output timescales is well-known in the field of machine learning (we refer readers with a
machine learning or data science background to Gauch and Lin (2020) for a general introduction to rainfall–runoff modeling).
The architectural flexibility of recurrent neural models allows for approaches that jointly process the different timescales in a
hierarchical fashion. Techniques to "divide and conquer" long sequences through hierarchical processing date back decades
(e.g., Schmidhuber (1991), Mozer (1991)). More recently, Koutník et al. (2014) proposed a "clockwork" architecture that
partitions a recurrent neural network into layers with individual clock speeds, where each layer is updated at its own frequency.
This way, lower-frequency layers allow the network to learn longer-term dependencies. Even in such a hierarchical approach,
however, the high-frequency neurons must process the full time series, which makes training slow. Neil et al. (2016) extended
an LSTM to process irregularly sampled inputs by means of a time gate that only attends to the input at steps of a learned
frequency. This helps discriminate overlaid input signals, but is likely unsuited to rainfall–runoff prediction because it has
no means of aggregating inputs while the time gate is closed. Chung et al. (2016) demonstrated how hierarchical processing
helps LSTMs to translate sentences and recognize handwriting, but the approach depends on a binary decision that is only
differentiable through a workaround.

Most of these models were designed for tasks like natural language processing and other non-physical applications. Unlike
these tasks, time series in rainfall–runoff modeling have regular frequencies with fixed translation factors (e.g., one day has al-
ways 24 hours), whereas words in natural language or strokes in handwriting vary in their length. The application area of Araya
et al. (2019) was closer in this respect—they predicted wind speed given input data at multiple timescales. But, like the afore-
mentioned language and handwriting applications, the objective was to predict a single time series—be it sentences, strokes, or
wind speed. Our objective encompasses *multiple* outputs, one for each target timescale. Hence, multi-timescale rainfall–runoff
prediction has similarities to multi-objective optimization. In our case, the different objectives are closely related, since aggre-
gation of discharge across time steps should be conservatory: For instance, every 24 hourly prediction steps should average (or
sum, depending on the units) to one daily prediction step. Rather than viewing the problem from a multi-objective perspec-
tive, Lechner and Hasani (2020) modeled time-*continuous* functions with ODE-LSTMs, a method that combines LSTMs with
a mixture of ordinary differential equations and recurrent neural networks (ODE-RNN). The resulting models can generate
continuous predictions at arbitrary granularity. Initially, this seems like a highly promising approach, however, it has several
drawbacks in our application: First, since one model generates predictions for all timescales, one cannot easily use different
forcings products for different target timescales. Second, ODE-LSTMs were originally intended for scenarios where the input
data arrives in *irregular* intervals. In our context, the opposite is true: the meteorological forcings have fixed frequencies and
are therefore highly regular. Also, for practical purposes we do not actually need predictions at arbitrary granularity—a fixed

---

[1]https://water.noaa.gov/about/nwm



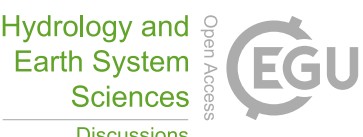

set of target timescales is sufficient. Lastly, in our exploratory experiments, using ODE-LSTMs to predict at timescales that were not part of training was actually worse (and much slower) than (dis-)aggregating the fixed-timescale predictions of our

multi-timescale LSTM. For these reasons, we excluded ODE-LSTMs from the main evaluation in this study (nevertheless, Appendix C details some of our exploratory ODE-LSTM experiments).

In this paper, we show how LSTM-based architectures can jointly predict discharge at multiple timescales in one model. We make the following contributions:

- First, we outline two LSTM architectures that predict discharge at multiple timescales (Sect. 2.3). We capitalize on
the fact that watersheds are damped systems: while the history of total mass and energy inputs are important, the impact of high-frequency variation becomes less important at long lead times. Our approach to providing multiple output timescales processes long-past input data at coarser temporal resolution than recent time steps. This shortens the input sequences, since high-resolution inputs are only necessary for the last few time steps. We benchmark their daily and hourly predictions against (i) a naive solution that trains individual LSTMs for each timescale and (ii) a traditional hydrologic
model, the US National Water Model (Sect. 3.1). Our results show that all LSTM solutions predict at significantly higher Nash–Sutcliffe efficiency than NWM at all timescales. While there is only little accuracy difference among the LSTMs, the naive model has much higher computational overhead than multi-timescale LSTMs.

- Second, we propose a regularization scheme that reduces inconsistencies across timescales as they arise from naive, per-timescale prediction (Sect. 2.3.2). According to our results, the regularization not only reduces inconsistencies but
also results in slightly improved predictions overall (Sect. 3.2).

- Third, we demonstrate that a multi-timescale LSTM can ingest individual and multiple sets of forcings for each target timescale, which closely resembles operational forecasting use-cases where forcings with high temporal resolution often have shorter lead times than forcings with low resolution. (Sect. 3.4).

## 2   Data and Methods

### 2.1   Data

In order to maintain some degree of continuity and comparability, we conducted our experiments in a way that is as comparable as possible with previous benchmarking studies (Newman et al., 2017; Kratzert et al., 2019) on the CAMELS dataset (Addor et al., 2017). Out of the 531 CAMELS basins used by previous studies, 516 basins have hourly stream gauge data available from the USGS Water Information System through the Instantaneous Values REST API.[2] This service provides historical mea-

surements at varying sub-daily resolutions (usually every 15 to 60 minutes), which we averaged to hourly and daily time steps for each basin. Since our forcing data and benchmark model data use UTC timestamps, we converted the USGS streamflow timestamps to UTC.

---

[2]https://waterservices.usgs.gov/rest/IV-Service.html



**Table 1.** NLDAS forcing variables used in this study (Xia et al., 2012).

| Variable | Units |
| --- | --- |
| Total precipitation | $\mathrm{kg\,m^{-2}}$ |
| Air Temperature | K |
| Surface pressure | Pa |
| Surface downward longwave radiation | $\mathrm{W\,m^{-2}}$ |
| Surface downward shortwave radiation | $\mathrm{W\,m^{-2}}$ |
| Specific humidity | $\mathrm{kg\,kg^{-1}}$ |
| Potential energy | $\mathrm{J\,kg^{-1}}$ |
| Potential evaporation | $\mathrm{J\,kg^{-1}}$ |
| Convective fraction | – |
| $u$ wind component | $\mathrm{m\,s^{-1}}$ |
| $v$ wind component | $\mathrm{m\,s^{-1}}$ |

While CAMELS provides only daily meteorological forcing data, we needed hourly forcings. To maintain some congruence with previous CAMELS experiments, we used the hourly NLDAS-2 product, which contains meteorological data since 1979 (Xia et al., 2012). We spatially averaged the forcing variables listed in Table 1 for each basin. Additionally, we averaged these basin-specific hourly meteorological variables to daily values.

We trained our models on the period from October 1, 1990 to September 30, 2003. The period from October 1, 2003 to September 30, 2008 was used as a validation period, where we evaluated different architectures and selected suitable model hyperparameters. Finally, we tested our models on data from October 1, 2008 to September 30, 2018.

All LSTM models used in this study take as inputs the eleven forcing variables listed in Table 1, concatenated at each time step with the same 27 static catchment attributes from the CAMELS dataset that Kratzert et al. (2019) used.

## 2.2 Benchmark Models

We used two groups of models as baselines for comparison with the proposed architectures: the LSTM proposed by Kratzert et al. (2019), naively adapted to hourly streamflow modeling, and the US National Water Model (NWM), a traditional hydrologic model used operationally by the National Oceanic and Atmospheric Administration.[3]

### 2.2.1 Traditional Hydrologic Model

The US National Oceanic and Atmospheric Agency (NOAA) generates hourly streamflow predictions with the National Water Model, which is a process-based model based on WRF-Hydro (Gochis et al., 2020). Specifically, we benchmarked against the NWM v2 reanalysis product, which includes hourly streamflow predictions for the years 1993 through 2018.

---

[3]https://water.noaa.gov/about/nwm

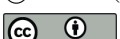



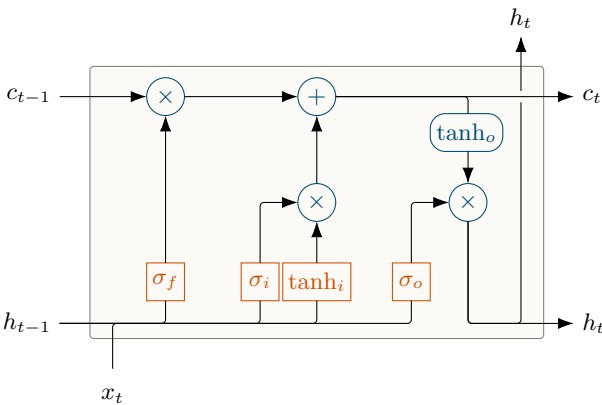

**Figure 1.** Schematic architecture of an LSTM cell with input $x_t$, cell state $c_t$, output $h_t$, and activations for forget gate ($\sigma_f$), input gate ($\sigma_i$, $\tanh_i$), and output gate ($\sigma_o$, $\tanh_o$). (Illustration derived from Olah (2015))

### 2.2.2 Naive LSTM

Long Short-Term memory (LSTM) networks (Hochreiter and Schmidhuber, 1997) are a flavor of recurrent neural networks designed to model long-term dependencies between input and output data. LSTMs maintain an internal memory state which is updated at each time step by a set of activated functions called *gates*. These gates control the input–state relationship (through the *input gate*), the state–output relationship (through the *output gate*), and the memory timescales (through the *forget gate*). Figure 1 illustrates this architecture. For a more detailed description of LSTMs, especially in the context of rainfall–runoff modeling, we refer to Kratzert et al. (2018).

LSTMs can cope with longer time series than classic recurrent neural networks because they are not susceptible to vanishing gradients during the training procedure (Bengio et al., 1994; Hochreiter and Schmidhuber, 1997). Since LSTMs process input sequences sequentially, longer time series result in longer training and inference time. For daily predictions, this is not a problem, since look-back windows of 365 days appear to be sufficient for most basins, at least in the CAMELS dataset. Therefore, our baseline for daily predictions is the LSTM model from Kratzert et al. (2019), which was trained on daily data with an input sequence length of 365 days.

For hourly data, even half of a year corresponds to more than 4300 time steps, which results in very long training and inference runtime, as we will detail in Sect. 3.3. In addition to the computational overhead, the LSTM forget gate makes it hard to learn long-term dependencies, because it effectively reintroduces vanishing gradients into the LSTM (Jozefowicz et al., 2015). Yet, we cannot simply remove the forget gate—both empirical LSTM analyses (Jozefowicz et al., 2015; Greff et al., 2017) and our exploratory experiments showed that this deteriorates results. To address this, Gers et al. (1999) proposed to initialize the bias of the forget gate to a small positive value (we used 3). This starts training with an open gate and enables gradient flow across more time steps.




We used this bias initialization trick for all our LSTM models, and it allowed us to include the LSTM with hourly inputs as the naive hourly baseline for our proposed models. The architecture for this naive benchmark is identical to the daily LSTM, except that we ingested input sequences of 4320 hours (180 days). Further, we tuned the learning rate and batch size for the naive hourly LSTM, since it receives 24 times the amount of samples than the daily LSTM. The extremely slow training impedes a more extensive hyperparameter search. Appendix D details the grid of hyperparameters we evaluated to find a

suitable configuration, as well as further details on the final hyperparameters.

## 2.3 Using LSTMs to Predict Multiple Timescales

We evaluated two different LSTM architectures that are capable of simultaneous predictions at multiple timescales. For the sake of simplicity, the following explanations use the example of a two-timescale model that generates daily and hourly predictions. Nevertheless, the architectures we describe here generalize to other timescales and to more than two timescales, as we will

show in an experiment in Sect. 3.5.

The first model, shared multi-timescale LSTM (sMTS-LSTM), is a simple extension of the naive approach: We generate a daily prediction as usual—the LSTM ingests an input sequence of $T_D$ time steps at daily resolution and outputs a prediction at the last time step (i.e., sequence-to-one prediction). Next, we reset the hidden and cell states to their values from time step $T_D - T_H/24$ and ingest the hourly input sequence of length $T_H$ to generate 24 hourly predictions that correspond to the last

daily prediction. In other words, during each prediction step, we perform two forward passes through the same LSTM: one that generates a daily prediction and one that generates 24 corresponding hourly predictions. We add a one-hot timescale encoding to the input sequence such that the LSTM can distinguish daily from hourly inputs. The key insight with this model is that the hourly forward pass starts with LSTM states from the daily forward pass, which act as a summary of long-term information up to that point. In effect, the LSTM has access to a large look-back window but, unlike the naive hourly LSTM, it does not suffer

from the performance impact of extremely long input sequences.

The second architecture, illustrated in Fig. 2, is a more general variant of the sMTS-LSTM that is specifically built for multi-timescale predictions, hence, we call it the *multi-timescale LSTM* (MTS-LSTM). It works just like the shared version but splits the LSTM into two branches, one for each timescale. We first generate a prediction with an LSTM acting at the coarsest timescale (here: daily) using a full input sequence of length $T_D$ (e.g., 365 days). Next, we reuse the daily hidden and cell states

from step $T_D - T_H/24$ as the initial states for an LSTM at a finer timescale (here: hourly), which generates the corresponding 24 hourly predictions. Since the two LSTM branches may have different hidden sizes, we feed the states through a linear state transfer layer ($\text{FC}_h, \text{FC}_c$) before reusing them as initial hourly states. In this setup, each LSTM branch only receives inputs of its respective timescale, hence, we do not need to one-hot encode that timescale. This architecture makes it clear why we call the other variant "shared" MTS-LSTM: Effectively, the sMTS-LSTM is an ablation of the MTS-LSTM. Both variants have the

same architecture, but the weights of the sMTS-LSTM are shared across all per-timescale branches and its state transfer layers are identity operations.





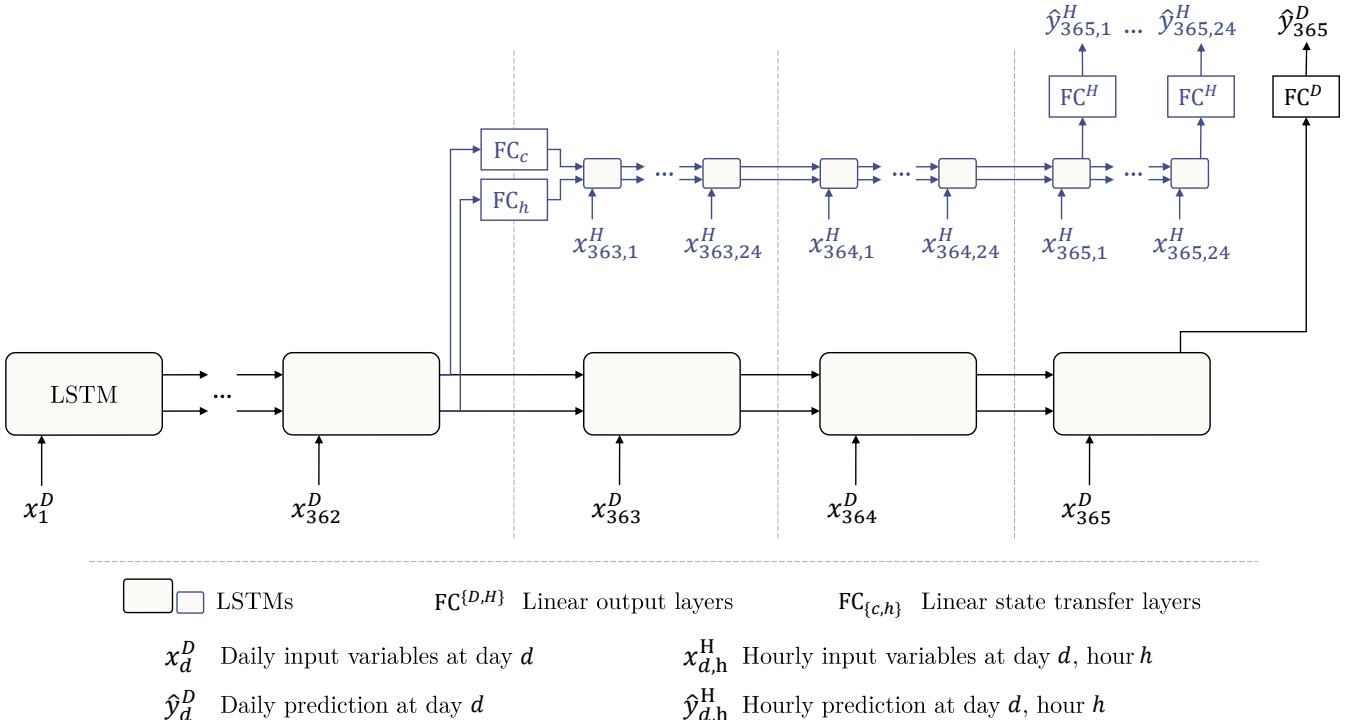

**Figure 2.** Illustration of the multi-timescale LSTM architecture that uses one distinct LSTM per timescale. In the depicted example, the daily and hourly input sequence lengths are $T^D = 365$ and $T^H = 72$. In the sMTS-LSTM model (i.e., without distinct LSTM branches), $FC_c$ and $FC_h$ are identity operations, and the two branches (including the fully-connected output layers $FC^H$ and $FC^D$) share their model weights.

### 2.3.1 Per-Timescale Input Variables

An important advantage of the MTS-LSTM over the sMTS-LSTM architecture arises from its more flexible input dimensionality. As each timescale is processed in an individual LSTM branch, we can ingest different input variables to predict the different timescales. This can be a key differentiator in operational applications when, for instance, there exist daily weather forecasts with a much longer lead time than the available hourly forecasts, or when using remote sensing data that is available only at certain overpass frequencies. In these cases, the MTS-LSTM can process the daily forcings in its daily LSTM branch and the hourly forcings in its hourly LSTM branch. As such, this per-timescale forcings strategy allows for using different inputs at different timescales.

To evaluate this capability, we used two sets of forcings as daily inputs: the Daymet and Maurer forcing sets that are included in the CAMELS dataset (Newman et al., 2014). For lack of other hourly forcing products, we conducted two experiments: in one, we continued to use only the hourly NLDAS forcings. In the other, we additionally ingested the corresponding day's Daymet and Maurer forcings at each hour. Since the Maurer forcings only range until 2008, we conducted this experiment on the validation period from October 2003 to September 2008.





### 2.3.2 Cross-Timescale Consistency

Since the MTS-LSTM and sMTS-LSTM architectures generate predictions at multiple timescales simultaneously, we can incentivize predictions that are consistent across timescales. Unlike in other domains (e.g., computer vision, Zamir et al. (2020)), consistency is well-defined in our application: predictions are consistent if the mean of every day's hourly predictions is the same as that day's daily prediction. Hence, we can explicitly state this constraint as a regularization term in our loss function. Building on the basin-averaged NSE loss from Kratzert et al. (2019), our loss function averages the losses from each individual timescale and regularizes with the mean squared difference between daily and day-averaged hourly predictions. Note that, although we describe the regularization with only two simultaneously predicted timescales, the approach generalizes to more timescales, as we can add the mean squared difference between each pair of timescale $\tau$ and next-finer timescale $\tau'$. All taken together, we used the following loss for a two-timescale daily–hourly model:

$$\mathrm{NSE}_{\mathrm{reg}}^{D,H} = \frac{1}{2} \underbrace{\sum_{\tau \in \{D,H\}} \left( \frac{1}{B} \sum_{b=1}^{B} \sum_{t=1}^{N_b^\tau} \frac{(\hat{y}_t^\tau - y_t^\tau)^2}{(\sigma_b + \epsilon)^2} \right)}_{\text{per-timescale NSE}} + \underbrace{\frac{1}{B} \sum_{b=1}^{B} \frac{1}{N_b^D} \sum_{t=1}^{N_b^D} \left( \hat{y}_t^D - \frac{1}{24} \sum_{h=1}^{24} \hat{y}_{t,h}^H \right)^2}_{\text{mean squared difference regularization}} \tag{1}$$

Here, $B$ is the number of basins, $N_b^\tau$ is the number of samples for basin b at timescale $\tau$, $y_t^\tau$ and $\hat{y}_t^\tau$ are observed and predicted discharge values for the $t$th time step of basin $b$ at timescale $\tau$. $\sigma_b$ is the observed discharge variance of basin $b$ over the whole training period, and $\epsilon$ is a small value that guarantees numeric stability. We predicted discharge in the same unit $(\mathrm{mm\,h^{-1}})$ at all timescales, so that the daily prediction is compared with an average across 24 hours.

### 2.3.3 Predicting More Timescales

While all our experiments so far considered daily and hourly input and output data, the MTS-LSTM architecture generalizes to other timescales. We showed this capability in a setup similar to the operational National Water Model,[4] albeit in a reanalysis setting: We trained an MTS-LSTM to predict the last 18 hours (hourly), 10 days (three-hourly), and 30 days (six-hourly) of discharge—all within one MTS-LSTM. Table 2 details the sizes of the input and output sequences for each timescale. To achieve a sufficiently long look-back window without using exceedingly long input sequences, we additionally predicted one day of daily streamflow but did not evaluate these daily predictions. In this setup, the quality of predictions remained high across all target timescales, as our results in Sect. 3.5 show.

Related to the selection of target timescales, a practical note: The state transfer from lower to higher timescales requires careful coordination of timescales and input sequence lengths. To illustrate, consider a toy setup that predicts two- and three-hourly discharge. Each timescale uses an input sequence length of 20 steps (i.e., 40 and 60 hours, respectively). The initial two-hourly LSTM state should then be the $(60 - 40)/3 = 6.67$th three-hourly state—in other words, the step widths are asynchronous at the point in time where the LSTM branches split. Of course, one could simply select the 6th or 7th step, but then either two hours remain unprocessed or one hour is processed twice (once in the three- and once in the two-hourly LSTM branch).

---

[4]https://water.noaa.gov/about/nwm





**Table 2.** Input sequence length and prediction window for each predicted timescale. The daily input merely acts as a means to extend the look-back window to a full year without generating overly long input sequences; we do not evaluate the daily predictions.

| Timescale | Input sequence length | Prediction window |
|---|---|---|
| Hourly | 168 hours (= 7 days) | 18 hours |
| Three-hourly | 168 steps (= 21 days) | 80 steps (= 10 days) |
| Six-hourly | 360 steps (= 90 days) | 120 steps (= 30 days) |
| Daily | 365 days | 1 day |

Instead, we suggest selecting sequence lengths such that the LSTM splits at points where the timescales' steps align. In the above example, sequence lengths of 20 (three-hourly) and 21 (two-hourly) would align correctly: $(60 - 42)/3 = 6$.

## 2.4 Evaluation Criteria

Following previous studies that used the CAMELS datasets (Kratzert et al., 2020; Addor et al., 2018), we benchmarked our models with respect to the metrics and signatures listed in Table 3. Hydrologic signatures are statistics of hydrographs, and thus not natively a comparison between predicted and observed values. For these values, we calculated the Pearson correlation coefficient between the signatures of observed and predicted discharge across the 516 CAMELS basins used in this study.

To quantify the cross-timescale consistency of the different models, we calculated the root mean squared deviation between daily predictions and hourly predictions when aggregated to daily values:

$$\text{MSD}^{D,H} = \sqrt{\frac{1}{T}\sum_{t=1}^{T}\left(\hat{y}_t^D - \frac{1}{24}\sum_{h=1}^{24}\hat{y}_{t,h}^H\right)^2} \tag{2}$$

To account for randomness in LSTM weight initialization, all the LSTM results reported are calculated on hydrographs that result from averaging the predictions of ten independently trained LSTM models.

## 3 Results

### 3.1 Benchmarking

Table 4 compares the median test period evaluation metrics of the MTS-LSTM and sMTS-LSTM architectures with the benchmark naive LSTM models and the process-based National Water Model (NWM). Figure 3 illustrates the cumulative distributions of per-basin NSE values of the different models. By this metric, all LSTM models, even the naive ones, outperform NWM at both hourly and daily time steps. All models perform slightly worse on hourly predictions than on daily predictions. Accordingly, the results in Table 4 list differences in median NSE values between NWM and the LSTM models ranging from 0.11 to 0.16 (daily) and around 0.19 (hourly). Among the LSTM models, the differences in all metrics are comparatively small. The sMTS-LSTM achieves the best daily and hourly median NSE at both timescales. The naive models produce NSE values that





**Table 3.** Evaluation metrics (upper table section) and hydrologic signatures (lower table section) used in this study. For each signature, we calculated the Pearson correlation between the signatures of observed and predicted discharge across all basins. Descriptions of the signatures are taken from Addor et al. (2018).

| Metric/Signature | Description | Reference |
|---|---|---|
| NSE | Nash–Sutcliffe efficiency | Eq. 3 in Nash and Sutcliffe (1970) |
| KGE | Kling–Gupta efficiency | Eq. 9 in Gupta et al. (2009) |
| Pearson r | Pearson correlation between observed and simulated flow | |
| $\alpha$-NSE | Ratio of standard deviations of observed and simulated flow | From Eq. 4 in Gupta et al. (2009) |
| $\beta$-NSE | Ratio of the means of observed and simulated flow | From Eq. 10 in Gupta et al. (2009) |
| FHV | Top 2% peak flow bias | Eq. A3 in Yilmaz et al. (2008) |
| FLV | Bottom 30% low flow bias | Eq. A4 in Yilmaz et al. (2008) |
| FMS | Bias of the slope of the flow duration curve between the 20% and 80% percentile | Eq. A2 Yilmaz et al. (2008) |
| Peak-timing | Mean time lag between observed and simulated peaks | See Appendix A |
| Baseflow index | Ratio of mean daily/hourly baseflow to mean daily/hourly discharge | Ladson et al. (2013) |
| HFD mean | Mean half-flow date (date on which the cumulative discharge since October first reaches half of the annual discharge) | Court (1962) |
| High flow dur. | Average duration of high-flow events (number of consecutive steps > 9 times the median daily/hourly flow) | Clausen and Biggs (2000), Table 2 in Westerberg and McMillan (2015) |
| High flow freq. | Frequency of high-flow days/hours (> 9 times the median daily/hourly flow) | Clausen and Biggs (2000), Table 2 in Westerberg and McMillan (2015) |
| Low flow dur. | Average duration of low-flow events (number of consecutive days/hours < 0.2 times the mean flow) | Olden and Poff (2003), Table 2 in Westerberg and McMillan (2015) |
| Low flow freq. | Frequency of low-flow days/hours (< 0.2 times the mean daily/hourly flow) | Olden and Poff (2003), Table 2 in Westerberg and McMillan (2015) |
| Q5 | 5% flow quantile (low flow) | |
| Q95 | 95% flow quantile (high flow) | |
| Q mean | Mean daily/hourly discharge | |
| Runoff ratio | Runoff ratio (ratio of mean daily/hourly discharge to mean daily/hourly precipitation, using NLDAS precipitation) | Eq. 2 in Sawicz et al. (2011) |
| Slope FDC | Slope of the flow duration curve (between the log-transformed 33rd and 66th streamflow percentiles) | Eq. 3 in Sawicz et al. (2011) |
| Stream elasticity | Streamflow precipitation elasticity (sensitivity of streamflow to changes in precipitation at the annual time scale, using NLDAS precipitation) | Eq. 7 in Sankarasubramanian et al. (2001) |
| Zero flow freq. | Frequency of days/hours with zero discharge | |

**Figure 3.** Cumulative NSE distributions of the different models. The top left plot (solid lines) shows NSEs of daily predictions, the top right plot (dashed lines) shows NSEs of hourly predictions, and the lower plot combines both plots into one view for comparison.

are slightly lower, but the difference is not statistically significant at $\alpha = 0.001$ (Wilcoxon signed-rank test: $p = 2.7 \times 10^{-3}$ (hourly), $p = 8.7 \times 10^{-2}$ (daily)). The NSE distribution of the MTS-LSTM is significantly different from the sMTS-LSTM ($p = 3.7 \times 10^{-19}$ (hourly), $p = 3.2 \times 10^{-29}$ (daily)), but the absolute difference is small. We leave it to the judgment of the reader to decide whether or not this difference is negligible from a hydrologic standpoint.

Beyond NSE, all LSTM models exhibit lower peak-timing errors than NWM. For hourly predictions, the median peak-timing error of the sMTS-LSTM is around three and a half hours, compared to more than six hours for NWM. Naturally, the peak-timing errors for daily predictions are smaller values since the error is measured in days. Consequently, the sMTS-LSTM yields a peak-timing error of 0.3 days, versus 0.5 days for NWM. The process-based NWM, in turn, often produces results with better flow bias metrics, especially with respect to low flows (FLV). This observation underscores conclusions from prior



**Figure 4.** NSE and peak-timing error by basin for daily and hourly sMTS-LSTM predictions. Brighter colors correspond to better values. Note the different color scales for daily and hourly peak-timing error.

work that indicate there is improvement to be made in arid climate regimes (Kratzert et al., 2020; Frame et al., 2020). Similarly,
the lower section of Table 4 lists correlations between hydrologic signatures of observed and predicted discharge: NWM has
the highest correlations for frequencies of high flows, low flows, and zero-flows, as well as for flow duration curve slopes.

Figure 4 visualizes the distributions of NSE and peak-timing error across space for the hourly predictions with the sMTS-
LSTM. As in previous studies, the NSE values are lowest in arid basins of the Great Plains and Southwestern United States.
The peak-timing error shows similar spatial patterns, however, especially the hourly peak-timing error also shows higher values
along the southeastern coastline.





**Table 4.** Median metrics (upper table section) and Pearson correlation between signatures of observed and predicted discharge (lower table section) across all 516 basins for the sMTS-LSTM, MTS-LSTM, naive daily and hourly LSTMs, and the process-based NWM. Bold values highlight results that are not significantly different from the best model in the respective metric or signature ($\alpha = 0.001$). See Table 3 for a description of the metrics and signatures.

| | daily | | | | hourly | | | |
|---|---|---|---|---|---|---|---|---|
| | sMTS-LSTM | MTS-LSTM | Naive | NWM | sMTS-LSTM | MTS-LSTM | Naive | NWM |
| NSE | **0.762** | 0.750 | **0.755** | 0.636 | **0.752** | 0.748 | **0.751** | 0.559 |
| MSE | **0.002** | 0.002 | **0.002** | 0.004 | **0.003** | 0.003 | **0.003** | 0.005 |
| RMSE | **0.048** | 0.049 | **0.048** | 0.059 | **0.054** | 0.055 | **0.054** | 0.071 |
| KGE | 0.727 | 0.714 | **0.760** | 0.666 | **0.731** | 0.726 | **0.739** | 0.638 |
| $\alpha$-NSE | 0.819 | 0.813 | **0.873** | **0.847** | 0.828 | 0.825 | **0.837** | **0.846** |
| Pearson r | **0.891** | 0.882 | 0.885 | 0.821 | **0.885** | 0.882 | 0.882 | 0.779 |
| $\beta$-NSE | **−0.055** | **−0.043** | **−0.042** | **−0.038** | **−0.045** | **−0.039** | **−0.039** | **−0.034** |
| FHV | −17.656 | −17.834 | **−13.336** | **−15.053** | −16.296 | −16.115 | **−14.467** | **−14.174** |
| FMS | −9.025 | −13.421 | −10.273 | **−5.099** | −9.274 | −12.772 | −8.896 | **−5.264** |
| FLV | **9.617** | **9.730** | 12.195 | 0.775 | −35.214 | −35.354 | −35.097 | **6.315** |
| Peak-timing | **0.306** | 0.333 | **0.310** | 0.474 | **3.540** | 3.757 | 3.754 | 5.957 |
| NSE (mean) | 0.662 | 0.602 | 0.631 | 0.471 | 0.652 | 0.620 | 0.644 | 0.364 |
| Number of NSEs < 0 | 10 | 12 | 18 | 37 | 13 | 17 | 13 | 46 |
| High-flow freq. | 0.599 | 0.486 | 0.598 | **0.730** | 0.588 | 0.537 | 0.619 | **0.719** |
| High-flow dur. | **0.512** | 0.463 | **0.491** | 0.457 | **0.433** | 0.416 | **0.471** | **0.316** |
| Low-flow freq. | 0.774 | 0.657 | 0.774 | **0.796** | 0.764 | 0.697 | 0.782 | **0.789** |
| Low-flow dur. | **0.316** | 0.285 | 0.280 | 0.303 | **0.309** | 0.274 | 0.307 | 0.160 |
| Zero-flow freq. | **0.392** | **0.286** | **0.409** | **0.502** | 0.363 | 0.401 | 0.493 | **0.505** |
| Q95 | 0.979 | 0.978 | **0.980** | 0.956 | 0.980 | 0.979 | 0.979 | **0.956** |
| Q5 | **0.970** | 0.945 | **0.979** | 0.928 | **0.968** | 0.955 | **0.964** | 0.927 |
| Q mean | 0.985 | 0.984 | **0.986** | 0.972 | 0.984 | 0.983 | 0.983 | 0.970 |
| HFD mean | 0.930 | **0.943** | **0.945** | 0.908 | **0.944** | **0.948** | 0.941 | 0.907 |
| Slope FDC | 0.556 | 0.430 | **0.679** | 0.663 | 0.635 | 0.633 | 0.647 | **0.712** |
| Stream elasticity | **0.601** | 0.560 | **0.615** | 0.537 | **0.601** | 0.563 | **0.626** | 0.588 |
| Runoff ratio | 0.960 | **0.957** | **0.962** | 0.924 | 0.955 | 0.954 | 0.952 | **0.918** |
| Baseflow index | **0.897** | 0.818 | **0.904** | 0.865 | **0.935** | 0.908 | **0.932** | 0.869 |





**Table 5.** Median and maximum root mean squared deviation ($\mathrm{mm\,d^{-1}}$) between daily and day-aggregated hourly predictions for the sMTS-LSTM with and without regularization, compared with independent prediction through naive LSTMs. $p$ and $d$ denote the significance (Wilcoxon signed-rank test) and effect size (Cohen's $d$) of the difference to the inconsistencies of the regularized sMTS-LSTM.

|  | Median | Maximum | $p$ | $d$ |
|---|---|---|---|---|
| sMTS-LSTM | 0.376 | 1.670 | – | – |
| sMTS-LSTM (no regularization) | 0.398 | 2.176 | $1.49 \times 10^{-25}$ | 0.091 |
| Naive | 0.490 | 2.226 | $7.09 \times 10^{-72}$ | 0.389 |

### 3.2 Cross-Timescale Consistency

Since the (non-naive) LSTM-based models jointly predict discharge at multiple timescales, we can incentivize predictions that are consistent across timescales. As described in Sect. 2.3.2, this happens through a regularized NSE loss function that penalizes inconsistencies.

To gauge the effectiveness of this regularization, we compare inconsistencies between timescales in the best benchmark model, the sMTS-LSTM, with and without regularization. As a baseline, we also compare against the cross-timescale inconsistencies from two independent naive LSTMs. Table 5 lists the mean, median, and maximum root mean squared deviation between the daily predictions and the hourly predictions when aggregated to daily values. Without regularization, simultaneous prediction with the sMTS-LSTM yields smaller inconsistencies than the naive approach (i.e., separate LSTMs at each

timescale). Cross-timescale regularization further reduces inconsistencies and results in a median root mean squared deviation of $0.376\,\mathrm{mm\,d^{-1}}$.

Besides reducing inconsistencies, the regularization term appears to have a small but beneficial influence on the overall skill of the daily predictions: With regularization, the median NSE increases slightly from 0.755 to 0.762. Judging from the hyperparameter tuning results, this appears to be a systematic improvement (rather than a fluke), because for both sMTS-LSTM

and MTS-LSTM, at least the three best hyperparameter configurations use regularization.

### 3.3 Computational Efficiency

In addition to differences in accuracy, the different LSTM architectures have rather large differences in computational overhead, and therefore runtime. The naive hourly model must iterate through 4320 input sequence steps for each prediction it makes, whereas the MTS-LSTM and sMTS-LSTM only require 365 daily and 336 hourly steps. Consequently, where the naive hourly

LSTM takes more than one day to train on one NVIDIA V100 GPU, the MTS-LSTM and sMTS-LSTM take just over 6 (MTS-LSTM) and 8 hours (sMTS-LSTM).

Moreover, while training is a one-time effort, the runtime advantage is even larger during inference: The naive model requires around 9 hours runtime to predict 10 years of hourly data for 516 basins on an NVIDIA V100 GPU. This is about 40 times





slower than the MTS-LSTM and sMTS-LSTM models, which both require around 13 minutes for the same task on the same
hardware—and the multi-timescale models generate daily predictions in addition to hourly ones.

### 3.4    Per-Timescale Input Variables

While the MTS-LSTM yields slightly worse predictions than the sMTS-LSTM in our benchmark evaluation, it has the important ability to ingest different input variables at each timescale. The following two experiments show how harnessing this feature can increase the accuracy of the MTS-LSTM beyond its shared version. In the first experiment, we used two daily input
forcing sets (Daymet and Maurer) and one hourly forcing set (NLDAS). In the second experiment, we additionally ingested the daily forcings into the hourly LSTM branch.

   Table 6 compares the results of these two multi-forcings experiments with the single-forcing models (MTS-LSTM and sMTS-LSTM) from the benchmarking section. The second experiment—ingesting daily inputs into the hourly LSTM branch—yields the best results. The additional daily forcings increase the median daily NSE by $0.045$—from $0.766$ to $0.811$. Even
though the hourly LSTM branch only obtains low-resolution additional values, the hourly NSE increases by $0.036$—from $0.776$ to $0.812$. This multi-input version of the MTS-LSTM is the best model in this study, significantly better than the best single-forcing model (sMTS-LSTM). An interesting observation is that it is *daily* inputs to the *hourly* LSTM branch that improve predictions. Using only hourly NLDAS forcings in the hourly branch, the hourly median NSE drops to $0.781$. Following Kratzert et al. (2020), we expect that additional hourly forcings datasets will have further positive impact on the hourly accuracy
(beyond the improvement we see with additional daily forcings).

### 3.5    Predicting More Timescales

In the above experiments, we evaluated our models on daily and hourly predictions only. The MTS-LSTM architectures, however, generalize to other timescales.

   Table 7 lists the NSE values on the test period for each timescale. To calculate metrics, we only consider the first eight
three-hourly and four six-hourly predictions that were made on each day. The hourly median NSE of $0.747$ is barely different from the median NSE of the daily–hourly MTS-LSTM ($0.748$). While the three-hourly predictions are roughly as accurate as the hourly predictions, the six-hourly predictions are slightly worse (median NSE $0.734$).

## 4    Discussion and Conclusion

The purpose of this work was to generalize LSTM-based rainfall–runoff modeling to multiple timescales. This task is not
as trivial as simply running different deep learning models at different timescales due to long look-back periods, associated memory leaks, and computational expense. With MTS-LSTM and sMTS-LSTM, we propose two LSTM-based rainfall–runoff models that make use of the specific (physical) nature of the simulation problem.

   The results show that the advantage LSTMs have over process-based models on daily predictions extends to sub-daily predictions. An architecturally simple approach, what we here call the sMTS-LSTM, can process long-term dependencies at





**Table 6.** Median validation metrics across all 516 basins for the MTS-LSTMs trained on multiple sets of forcings (multi-forcing A uses daily Daymet and Maurer forcings as additional inputs into the hourly models, multi-forcing B uses just NLDAS as inputs into the hourly models). For comparison, the table shows the results for the single-forcing MTS-LSTM and sMTS-LSTM. Bold values highlight results that are not significantly different from the best model in the respective metric ($\alpha = 0.001$).

|  | daily | | | | hourly | | | |
|---|---|---|---|---|---|---|---|---|
|  | multi-forcing | | single-forcing | | multi-forcing | | single-forcing | |
|  | A | B | sMTS-LSTM | MTS-LSTM | A | B | sMTS-LSTM | MTS-LSTM |
| NSE | **0.811** | **0.805** | 0.785 | 0.766 | **0.812** | 0.781 | 0.783 | 0.776 |
| MSE | **0.002** | **0.002** | 0.002 | 0.002 | **0.002** | 0.002 | 0.002 | 0.002 |
| RMSE | **0.040** | **0.039** | 0.043 | 0.042 | **0.045** | 0.049 | 0.048 | 0.049 |
| KGE | **0.782** | 0.777 | **0.779** | 0.760 | **0.801** | 0.788 | 0.779 | 0.768 |
| $\alpha$-NSE | **0.879** | 0.874 | 0.865 | 0.853 | **0.905** | 0.888 | 0.886 | 0.874 |
| Pearson r | **0.912** | **0.911** | 0.902 | 0.895 | **0.911** | 0.898 | 0.901 | 0.895 |
| $\beta$-NSE | 0.014 | 0.018 | −0.014 | **−0.007** | 0.014 | 0.007 | −0.006 | **−0.002** |
| FHV | **−10.993** | −11.870 | −13.562 | −14.042 | **−8.086** | −9.854 | −10.557 | −11.232 |
| FMS | −14.179 | −14.388 | **−9.336** | −13.085 | −13.114 | −13.437 | **−10.606** | −12.996 |
| FLV | **14.034** | 12.850 | 9.931 | **14.486** | −26.969 | −27.567 | **−34.003** | −62.439 |
| Peak-timing | **0.250** | **0.273** | **0.286** | **0.308** | 3.846 | 3.711 | **3.571** | 3.800 |
| NSE (mean) | 0.661 | 0.663 | 0.669 | 0.603 | 0.679 | 0.630 | 0.657 | 0.615 |
| Number of NSEs < 0 | 18 | 16 | 13 | 15 | 17 | 21 | 20 | 20 |

**Table 7.** Median test period NSE and number of basins with NSE below zero across all 516 basins for NWM and the MTS-LSTM model trained to predict 1-, 3-, and 6-hourly discharge. The three- and six-hourly NWM results are for aggregated hourly predictions.

| Timescale | Model | Median NSE | Number of NSEs < 0 |
|---|---|---|---|
| Hourly | MTS-LSTM | 0.747 | 17 |
|  | NWM | 0.562 | 47 |
| Three-hourly | MTS-LSTM | 0.746 | 15 |
|  | NWM | 0.570 | 44 |
| Six-hourly | MTS-LSTM | 0.734 | 12 |
|  | NWM | 0.586 | 42 |

much smaller computational overhead than a naive hourly LSTM. Nevertheless, the high accuracy of the naive hourly model shows that LSTMs, even with a forget gate, can cope with very long input sequences. Additionally, the LSTMs produce hourly predictions that are almost as good as their daily predictions, while the NWM's accuracy drops significantly from daily to





hourly predictions. A more extensive hyperparameter tuning might even increase the accuracy of the naive model, however, this is difficult to test because of the large computational expense of training LSTMs with high-resolution input sequences that

are long enough to capture all hydrologically relevant history.

The high quality of the sMTS-LSTM model indicates that the "summary" state between daily and hourly LSTM components contains as much information as the naive model extracts from the full hourly input sequence. This is an intuitive assumption, since the informative content of high-resolution forcings diminishes as we look back farther.

The MTS-LSTM adds to that the ability to use distinct sets of input variables for each timescale. This is an important

operational use-case, as forcings with high temporal resolution often have shorter lead times than low-resolution products. In addition, per-timescale input variables allow for input data with lower temporal resolution, such as remote sensing products, without interpolation. Besides these conceptual considerations, this feature boosts model accuracy beyond the best single-forcings model, as we can ingest multiple forcing products at each timescale. The results from ingesting mixed input resolutions into the hourly LSTM branch (hourly NLDAS and daily Daymet and Maurer) highlight the flexibility of machine learning

models and show that daily forcing histories can contain enough information to support hourly predictions. Yet, there remain a number of steps to be taken before these models can be truly operational:

- First, like most academic rainfall–runoff models, our models operate in a reanalysis setting rather than performing actual lead-time forecasts.

- Second, operational models predict other hydrologic variables in addition to streamflow. Hence, multi-objective opti-

mization of LSTM water models is an open branch of future research.

- Third, the implementation we use in this paper carries out the predictions of more granular timescales only whenever it predicts a low-resolution step, where it predicts multiple steps to offset the lower frequency. For instance, at daily and hourly target timescales, the LSTM predicts 24 hourly steps *once a day*. In a reanalysis setting, this does not matter, but in a forecasting setting, one needs to generate hourly predictions more often than daily predictions. Note, however, that

this is merely a restriction of our implementation, not an architectural one: By allowing variable-length input sequences, we could produce one hourly prediction each hour (rather than 24 each day).

This work represents one step toward developing operational hydrologic models based on deep learning. Overall, we believe that the MTS-LSTM is the most promising model for future use. It can integrate forcings of different temporal resolutions, generates accurate and consistent predictions at multiple timescales, and its computational overhead both during training and

inference is far smaller than that of individual models per timescale.

*Code and data availability.* We trained all our machine learning models with the `neuralhydrology` Python library (https://github.com/ neuralhydrology/neuralhydrology). All code to reproduce our models and analyses is available at https://github.com/gauchm/mts-lstm. The trained models and their predictions are available at https://doi.org/10.5281/zenodo.4071885. Hourly NLDAS forcings and observed





**Appendix A: A Peak-Timing Error Metric**

Especially for predictions at high temporal resolutions, it is important that a model not only captures the correct magnitude of a flood event, but also its timing. We measure this with a "peak-timing" metric that quantifies the lag between observed and predicted peaks. First, we heuristically extract the most important peaks from the observed time series: starting with all observed peaks, we discard all peaks with topographic prominence smaller than the observed standard deviation and subsequently remove the smallest remaining peak until all peaks have a distance of at least 100 steps. Then, we search for the largest prediction within a window of one day (for hourly data) or three days (for daily data) around the observed peak. Given the pairs of observed and predicted peak time, the peak-timing error is their mean absolute difference.

**Appendix B: Negative Results**

Throughout our experiments, we found LSTMs to be a highly resilient architecture: We tried many different approaches to multi-timescale prediction, and a large fraction of them worked reasonably well, albeit not quite as well as the MTS-LSTM models we present in this paper. Nevertheless, we believe it makes sense to report some of these "negative" results—models that turned out not to work as well as the ones we finally settled on. Note, however, that the following reports are based on exploratory experiments with a few seeds and no extensive hyperparameter tuning.

**B1    Delta Prediction**

Extending our final MTS-LSTM, we tried to facilitate hourly predictions for the LSTM by ingesting the corresponding day's prediction into the hourly LSTM branch and only predicting each hour's deviation from the daily mean. If anything, however, this approach slightly deteriorated the prediction accuracy (and made the architecture more complicated).

We then experimented with predicting 24 weights for each day that distribute the daily streamflow across the 24 hours. This would have yielded the elegant side-effect of guaranteed consistency across timescales: the mean hourly prediction would always be equal to the daily prediction. Yet, the results were clearly worse, and, as we show in Sect. 3.2, we can achieve near-consistent results by incentive (regularization) rather than enforcement. One possible reason for the reduced accuracy is that it may be harder for the LSTM to learn two different things—predicting hourly weights and daily streamflow—than to predict the same streamflow at two timescales.

**B2    Cross-Timescale State Exchange**

Inspired by residual neural networks (ResNets) that use so-called skip connections to bypass layers of computation and allow for a better flow of gradients during training (He et al., 2016), we devised a "ResNet-multi-timescale LSTM" where after each



day, we combine the hidden state of the hourly LSTM branch with the hidden state of the daily branch into the initial daily and hourly hidden states for the next day. This way, we hoped, the daily LSTM branch might obtain more fine-grained information

about the last few hours than it could infer from its daily inputs. While the daily NSE remained roughly the same, the hourly predictions in this approach became much worse.

### B3   Multi-Timescale Input, Single-Timescale Output

For both sMTS-LSTM and MTS-LSTM, we experimented with ingesting both daily and hourly data into the models, but only training them to predict hourly discharge. In this setup, the models could fully focus on hourly predictions rather than trying

to satisfy two possibly conflicting goals. Interestingly, however, the hourly-only predictions were worse than combined multi-timescale predictions. One reason for this effect may be that the state summary that the daily LSTM branch passes to the hourly branch is worse, as the model obtains no training signal for its daily outputs.

### Appendix C:  Time-Continuous Prediction with ODE-LSTMs

As we already alluded to in the introduction, the combination of ordinary differential equations (ODEs) and LSTMs presents

another approach to multi-timescale prediction—one that is more accurately characterized as *time-continuous prediction* (as opposed to our multi-objective learning approach that predicts at arbitrary but fixed timescales). The ODE-LSTM passes each input time step through a normal LSTM, but then post-processes the resulting hidden state with an ODE that has its own learned weights. In effect, the ODE component can adjust the LSTM's hidden state to the time step size. For operational streamflow prediction, we believe that our MTS-LSTM approach is better suited, since ODE-LSTMs cannot directly process

different input variables for different target timescales. That said, from a scientific standpoint, we think that the idea of training a model that can then generalize to arbitrary granularity is of great interest (e.g., toward a more comprehensive and interpretable understanding of hydrologic processes).

Although the idea of time-continuous predictions seemed promising, in our exploratory experiments it was better to use an MTS-LSTM and aggregate (or dis-aggregate) its predictions to the desired target temporal resolution. Note that, due to the

slow training of ODE-LSTMs, we carried out the following experiments on ten basins (training one model per basin; we used smaller hidden sizes, higher learning rates, and trained for more epochs to adjust the LSTMs to this setting). Table C1 gives examples for predictions at untrained timescales: Table section (A) shows the mean and median NSE values across the ten basins when we trained the models on daily and 12-hourly target data but then generated hourly predictions (for the MTS-LSTM, we obtained hourly predictions by uniformly spreading the 12-hourly prediction across 12 hours). Table section (B)

shows the results when we trained on hourly and three-hourly target data but then predicted daily values (for the MTS-LSTM, we aggregated eight three-hourly predictions into one daily time step). These initial results show that, in almost all cases, it is better to (dis-)aggregate MTS-LSTM predictions than to use an ODE-LSTM.



**Table C1.** Test period NSE for the MTS-LSTM and ODE-LSTM models trained on two and evaluated on three target timescales. Section A: training on daily and 12-hourly data, evaluation additionally on hourly predictions (hourly MTS-LSTM results obtained as 12-hourly predictions uniformly distributed across 12 hours). Section B: training on hourly and three-hourly data, evaluation additionally on daily predictions (daily MTS-LSTM predictions obtained as averaged three-hourly values). The mean and median NSE are aggregated across the results for ten basins; best values are highlighted in bold.

| | Timescale | Timescale used in training loss | median NSE | | mean NSE | |
|---|---|---|---|---|---|---|
| | | | MTS-LSTM | ODE-LSTM | MTS-LSTM | ODE-LSTM |
| **(A)** | daily | yes | **0.726** | 0.720 | **0.664** | 0.651 |
| | 12-hourly | yes | **0.734** | 0.706 | **0.672** | 0.638 |
| | hourly | no | **0.706** | 0.639 | **0.634** | 0.592 |
| **(B)** | daily | no | **0.746** | 0.587 | **0.718** | 0.546 |
| | 3-hourly | yes | **0.728** | 0.675 | **0.672** | 0.593 |
| | hourly | yes | **0.700** | 0.677 | **0.633** | 0.586 |

## Appendix D: Hyperparameter Tuning

For the multi-timescale models, we performed a two-stage tuning. In the first stage, we trained architectural model parameters
(regularization, hidden size, sequence length, dropout) for 30 epochs at a batch size of 512 and a learning rate that starts at 0.001, reduces to 0.0005 after ten epochs, and to 0.0001 after 20 epochs. We selected the configuration with the best median NSE (we considered the average of daily and hourly median NSE) and, in the second stage, tuned its learning rate and batch size. Table D1 lists the parameter combinations we explored.

We did not tune architectural parameters of the naive LSTM models, since the architecture has already been extensively
tuned by Kratzert et al. (2019). The 24 times larger training set of the naive hourly model did, however, require additional tuning of learning rate and batch size, and we only trained for one epoch. As the extremely long input sequences greatly increase training time, we can only evaluate a relatively small parameter grid for the naive hourly model.

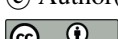



**Table D1.** Hyperparameter tuning grid. The upper table section lists the architectural parameters tuned in stage one, the lower section lists the parameters tuned in stage two. In both stages, we trained three seeds for each parameter combination and averaged their NSEs. Bold values denote the best hyperparameter combination for each model.

| | Naive (daily) | Naive (hourly) | sMTS-LSTM | MTS-LSTM |
|---|---|---|---|---|
| Loss | NSE | NSE | NSE | NSE |
| Regularization (see Sect. 2.3.2) | – | – | **yes**, no | **yes**, no |
| Hidden size | 256 | 256 | 64, **128**, 256, 512, 1024 | $2 \times 32$, $\mathbf{2 \times 64}$, $2 \times 128$, $2 \times 256$, $2 \times 512$ |
| Sequence length | 365 days | 4320 hours | **365 days** + 72, 168, **336 hours** | **365 days** + 72, 168, **336 hours** |
| Dropout | 0.4 | 0.4 | 0.2, **0.4**, 0.6 | 0.2, **0.4**, 0.6 |
| Learning rate[*] | (1: 5e-3, 10: 1e-3, 25: 5e-4) | 5e-4, **1e-4** | (1: 5e-3, 10: 1e-3, 25: 5e-4), (1: 1e-3, 10: 5e-4, 25: 1e-4), **(1: 5e-4, 10: 1e-4, 25: 5e-5)**, (1: 1e-4, 10: 5e-5, 25: 1e-5) | (1: 5e-3, 10: 1e-3, 25: 5e-4), (1: 1e-3, 10: 5e-4, 25: 1e-4), **(1: 5e-4, 10: 1e-4, 25: 5e-5)**, (1: 1e-4, 10: 5e-5, 25: 1e-5) |
| Batch size | 256 | **256**, 512 | **128**, 256, 512, 1024 | 128, **256**, 512, 1024 |
| Number of combinations | 1 | 4 | $90 + 16$ | $90 + 16$ |

[*] (1: $r_1$, 5: $r_2$, 10: $r_3$, …) denotes a learning rate of $r_1$ for epochs 1 to 4, of $r_2$ for epochs 5 to 10, etc.



*Author contributions.* MG, FK, and DK designed all experiments. MG conducted all experiments and analyzed the results, together with FK and DK. GN supervised the manuscript from the hydrologic perspective, JL and SH from the machine learning perspective. All authors 395 worked on the manuscript.

*Competing interests.* The authors declare that they have no conflict of interest.

*Acknowledgements.* This research was undertaken thanks in part to funding from the Canada First Research Excellence Fund and the Global Water Futures Program, and enabled by computational resources provided by Compute Ontario and Compute Canada. The ELLIS Unit Linz, the LIT AI Lab, and the Institute for Machine Learning are supported by the Federal State Upper Austria. We thank the projects AI-400 MOTION (LIT-2018-6-YOU-212), DeepToxGen (LIT-2017-3-YOU-003), AI-SNN (LIT-2018-6-YOU-214), DeepFlood (LIT-2019-8-YOU-213), Medical Cognitive Computing Center (MC3), PRIMAL (FFG-873979), S3AI (FFG-872172), DL for granular flow (FFG-871302), ELISE (H2020-ICT-2019-3 ID: 951847), AIDD (MSCA-ITN-2020 ID: 956832). Further, we thank Janssen Pharmaceutica, UCB Biopharma SRL, Merck Healthcare KGaA, Audi.JKU Deep Learning Center, TGW LOGISTICS GROUP GMBH, Silicon Austria Labs (SAL), FILL Gesellschaft mbH, Anyline GmbH, Google (Faculty Research Award), ZF Friedrichshafen AG, Robert Bosch GmbH, Software Competence 405 Center Hagenberg GmbH, TÜV Austria, and the NVIDIA Corporation.





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
