# Peer review of "Rainfall–Runoff Prediction at Multiple Timescales with a Single Long Short-Term Memory Network"

_Hydrology and Earth System Sciences, 2020_

## Referee Comment (RC1) · Jens Kiesel (Referee) · 14 Dec 2020

**GENERAL COMMENTS**

The manuscript "Rainfall–Runoff Prediction at Multiple Timescales with a Single Long Short-Term Memory Network" (LSTM) by Martin Gauch et al. presents an extension of LSTM hydrological models to sub-daily time steps. In previous publications, LSTMs as hydrological models were used on a daily time step. The authors explore multiple approaches to achieve a 'multi-timescale' model, of which three (naive LSTM, sMTS-LSTM, MTS-LSTM) are evaluated in more detail and less promising experiments are briefly explained in an Annexe. Similar to previous applications of LSTMs, the models

are applied at the CAMELS dataset, encompassing 516 basins across the contiguous USA where hourly data is available. Results are compared to the NOAA National Water Model (NWM) and show that all LSTMs architectures outperform the NWM. The authors suggest that the MTS-LSTM provides most flexbility for future use.

The manuscript is generally well written and structured, figures and tables support the results. Having a more process-based hydrological background, I nevertheless read the paper with interest and believe it fits well in the scope of HESS. I see the work as highly relevant, especially in the field of flood modelling and (eventually) forecasting, but also generally in the application of LSTMs at different temporal resolutions. However, especially regarding the latter, I think the authors should invest more work to improve the usefulness of the paper. Please find below more detailed comments, questions and suggestions that hopefully initiate a fruitful discussion and help in improving the paper.

SPECIFIC COMMENTS

ABSTRACT: I suggest to mention the difficulties and challenges applying the models (parameter estimation) and discuss the work still to be done regarding different time scales (e.g. generalization of parameters)

INTRODUCTION I think you are missing a research gap in your introduction which is important to apply the LSTM for different time steps, since there seems to be a time-step dependency of model parameters / hyperparameters (e.g. hidden size, sequence length, batch size, forget gate bias, learning rate?, others?). Due to the computationally expensive training of LSTMs, knowing which ones need to be adjusted, in about which range and identifying ideal values is essential. I would like to see this topic included in the "contributions" you list at the end of the introduction (and therefore also more prominently in the respective chapters).

p.2 l.29-41: I think this section is difficult to understand for a reader without firm neural networks background. Particularly phrases like: "partitions a recurrent neural network into layers with individual clock speeds", "process irregularly sampled inputs by means

of a time gate that only attends to the input at steps of a learned frequency", "the approach depends on a binary decision that is only differentiable through a workaround". I acknowledge that your paper cannot serve as an introduction to the topic. I have no clear suggestion other than making this paragraph more accessible to readers with a hydrological background through using less specialized jargon, if possible.

p.2 l.45 and 47: You write that Araya et al predicted wind speed at "multiple timescales". Then you mention that your objective is "multiple outputs, one for each target timescale". I don't understand the difference between that.

p.2 l.54ff: I see the capability to process input data in irregular intervals as an advantage. Think of satellite products that have different data gap length (e.g. soil moisture or altimetry products combining multiple sensors). You can discuss this further, but at least I suggest to write on p.3 l.77: "...LSTM can ingest individual and multiple sets of forcings each having regular time intervals for each target timescale. This closely resembles..."

p.3 l.70-72, 74-75: I suggest not to mention the results of your study in the introduction

p.3 l.64-78: These three paragraphs reveal that your introduction could be structured a bit better, ideally introducing the reader to these three problems/research gaps that need to be solved for "Rainfall–Runoff Prediction at Multiple Timescales with a Single Long Short-Term Memory Network". You have motivated the first paragraph, but the second and third 'contribution' that you list appears a bit unexpected since your previous introduction does not resemble that structure. For instance, instead of refering to sections later in the paper, I believe it would be better to introduce the reader to the problem of inconsistencies. You briefly mention this on p.2 l.27-28 for conventional hydrological models, but this can be extended, especially targeted on machine learning.

DATA AND METHODS

p.4 l.92-94: The distinction into training, validation and test is not fully clear to me. You

use the validation period to evaluate different architectures and to select model hyper-parameters. Could you elaborate on the reason why the evaluation of architecture and the hyperparameter selection cannot/should not be done during the training period?

p.4 l.101ff: Can you describe the datasets used in NWM and the basic characteristics (e.g. spatial application range, calibration strategy and performance) of the v2 reanalysis product?

p.5 Fig1: please also mention what "x" and "+" represent

p.6 l.127-130: I am particularly interested in how you tuned these parameters and how you decided which parameters to adjust and which ones not. As you mention, the LSTM application is computationally expensive and parameter selection and ranges are therefore important. Therefore, I would rather want to see Appendix D in the main text, and include information why certain parameters are time step dependent and others not. Also, In Table D1, it seems you ended up with 336 hrs sequence length for both architectures. Would an even longer sequence length lead to better results? What is the tradeoff between higher sequence lengths and computational costs?

p.6 l. 146-156: Could you explain why these two different LSTM architectures were developed? What are the expected advantages/disadvantages? The last sentence is crucial for the undestanding of the differences, I believe "weights of the sMTS-LSTM are shared across all per-timescale branches and its state transfer layers are identity operations." What is an identity operation?

p7. Figure 2: I understood from the text that both the sMTS-LSTM amd MTS-LSTM are branching out at each day into hourly predictions. The MTS-LSTM predicts 24 hours, using 72hrs sequence length. Is this the same for the sMTS-LSTM? The difference between sMTS-LSTM and MTS-LSTM is difficult to understand from just the figure caption. I think it would help to construct the illustration for both architectures to visualize the differences, if possible including the different weights for the MTS-LSTM and the similar weights for the sMTS-LSTM in the diagram.

p.7 l.158: I don't understand why the MTS-LSTM is more flexible in terms of input data than the sMTS-LSTM. In the sMTS-LSTM section you write (p.6 l.139): "we....ingest the hourly input sequence of length TH to generate 24 hourly predictions that correspond to the last daily prediction." Looking at Fig 2, to me this is similar in the MTS-LSTM, where the daily forcings have an effect until the hourly branch starts and then no update using the daily forcings/predictions seems to be made in the hourly branch. Therefore, effectively, you use the daily data until the model branches out and then you use the hourly forcings only? Again, I think it would help to show both architectures in Fig 2.

p.8 l.170-184: If I understand it correctly, adding the term into the loss function 'encourages' the model to minimize the difference between daily and sub-daily simulation. But similar to the NSE, this ideal value may not be reached, ending up with a model that is not consistent - even if you put an exceptionally high weight on the mean squared difference? Is there a reason why you don't 'force' consistency across timescales? E.g. when looking at Figure 2 I imagine you could add a function (e.g. simple multiplication of a term) that scales either the daily or the sub-daily prediction (or the average between the two) so that both match the consistency criteria (I now notice that may be similar to what you did in "B1 Delta Prediction")?

p.9 Table 2: it is a bit confusing to have these different sequence lengths. In the previous section it is 72hrs, here 168hrs, in Table D1 it is 336hrs. Can you harmonize this or explain why there are these differences?

RESULTS:

p.9 l.210: that means running ten seeds based on the parameterization in bold in Table D1? If so, I'd add this here

p.9 l.219: I find this particularly interesting when thinking about hydrological processes. The model parameter values (hidden and cell states) of the last coarse time step (Td - Th/24) are basically your boundary condition/initial state for the hourly model. It seems a bit counterintuitive that the sMTS-LSTM performs better than the naively trained full

hourly LSTM. So the 'error' you introduce through the daily average initial state must be insignificant (due to a sufficiently long sequence length?). Particularly in small basins and for flood peak prediction, this may not always be the case. A plot showing the spatial differences in performance between the naively trained LSTM, the sMTS-LSTM and MTS-LSTM (e.g. similar to Fig 4) could reveal if/where these differences exist. I'd however not be surprised if this plot will show no pattern due to input data uncertainty and randomness in the LSTM and the small performance difference between the LSTM types.

p.14 l.237-250: Interestingly, the Naive LSTM deviates most - probably because the sMTS-LSTM and the MTS-LSTM use recent states from the daily model and are therefore 'closer' to the daily models flow (volume) prediction? The beneficial influence on the NSE could arise because you are introducing a 'physically plausible' constraint in the model which 'helps' adapting the network to the processes? (see also my comment to p8. l.170-184). That is an interesting prospect and if true, could mean adding more of such physical constraints (e.g. global water balance closure) could improve the LSTM even further?

CONCLUSIONS:

p.16 l.292: it depends on how the NWM was calibrated and what the main purpose is (see also comment to p.4 l.101ff)

p.17 l.293: I understand and agree. But given that LSTMs perform so well for hydrological modelling, efforts should be made to generalize the hyperparameter values for different time steps. I believe you were not sufficiently confident with your tests to deduce general rules for the hyperparameter settings (and that may be a reason why this analysis ended up in the Annexe). But I think it would help the future application of LSTMs if you could give a summary of your experience: e.g. which parameters are time-step dependent, should a parameter increase or decrease with increasing/decreasing time steps, what if someone applies an even coarser time step (monthly)?

p.17 l.296-298: I know the differences are not statistically significant, but can you speculate on why the models are ranked in that order? Somehow the naive hourly LSTM seems not to be able to use this additional information content, or the half year sequence length is not suffcient to depict all states (e.g. groundwater storages may need longer sequence length in some catchments)?

p.17 l.299-305: Can you speculate why the daily forcings to the hourly MTS-LSTM improve the performance?

I believe there is more research to be done that you can mention here? E.g. a thorough investigation of time step-dependency of hyperparameters, find measures to use physical constraints in the LSTM (e.g. the regularization)

TECHNICAL CORRECTIONS

once introduced, you can stick to the abbreviations (e.g. NWM, MTS-LSTM)

p.1 l.14: LSTMs can predict hydrological processes in multiple...

p.3 l.58-60: I think you can refer to Appendix C here

p.5 l.118: ...half a year...

p.8 l.191-192: it is uncommon to mention results in the methods

p.8 l.199: this link is supplied here for the third time. Not sure if this is how HESS wants to have references to URLs.

p.9 l.215: 'even the naive ones' - the naive LSTM acts as a benchmark, so it is expected it performs better than (s)MTS?

p.9 l.216: I think it is fair to add that this worse performance on hourly is much more visible at the NWM

p.14 l.255: which parameterization and number of basins is meant here? I can't imagine you mean all basins, 10 seeds, 30 epochs?

p.17 l.311-312: I find this first sentence difficult to understand. If possible, split in two

p.18 l.334: I like the documentation of the failed approaches and where appropriate, I suggest to reference these in the main text

---

## Referee Comment (RC2) · Thomas Lees (Referee) · 3 Jan 2021

**1 OVERVIEW**

This paper seeks to answer the question: "Can a single LSTM model be used to produce accurate and consistent discharge simulations at daily timescales and sub-daily timescales?".

The major finding was that yes, you can use a single LSTM to produce daily and hourly predictions. Furthermore, compared with more traditional hydrological models, the MTS-LSTM shows a much smaller performance deterioration when comparing daily

simulations (better) to hourly simulations (only slightly worse).

The novel contributions of this paper are threefold:

1. The development of a new multi-timescale LSTM (MTS-LSTM) that produces discharge simulations at both daily and sub-daily timescales (including the flexibility to include arbitrary timescales).

2. The manipulation of the loss function to explicitly account for prior knowledge about the translation between daily and sub-daily timescales. Related to the "hierarchical" nature of these timescales.

3. The benchmarking of a suite of LSTM-based models against the operationally used NOAA National Water Model (NWM).

Research into LSTM based rainfall-runoff modelling has, thus far, mainly focused on simulations at daily timescales. This paper provides a welcome addition to the literature, since sub-daily trends can be important for flood impacts and for water resource managers. The authors focus on producing discharge simulations at a daily timescale and an hourly timescale, although they also show results for 3-hourly and 6-hourly timescales (see Table 2 p9, Table 7 p16).

In order to explore the LSTM architectures that can produce discharge simulations at multiple timescales, the authors suggest three possible avenues (more are included in Appendix B):

- Multiple LSTMs with different timescales, an hourly LSTM and a daily LSTM (naive).

- A "shared" multiple-timescale LSTM (sMTS-LSTM), which overcomes the problems of overly long input sequences, causing long training and inference times for the naive model.

- The MTS-LSTM, which overcomes the problems of the sMTS-LSTM being unable to include different input data for the different timescales.

Both the sMTS-LSTM and the MTS-LSTM are novel contributions to both hydrological modelling, and as far as I am aware, machine learning more generally. My main comment about the paper is that the difference between the sMTS-LSTM and the MTS-LSTM could be made clearer.

The authors describe four experiments to demonstrate the usefulness of their newly developed models:

1. Benchmark the MTS models (sMTS-LSTM & MTS-LSTM) against traditional hydrological models (NOAA NWM) and the naive LSTM (which although "naive" is still the most difficult benchmark to compete with). This comparison is thorough and explores accuracy across the hydrograph (see Table 4, Figure 3, Figure 4).

2. Explore the consistency of the MTS models hourly discharge predictions when aggregated to the models daily discharge predictions. The regularisation of the loss function improved the consistences of the sMTS-LSTM

3. Compare the computational efficiency of the 3 LSTM-based models. The MTS-LSTM was the most computationally efficient.

4. Test whether including the same information from different timescales improves model accuracy. The extra information improved forecast accuracy over a range of performance metrics (Table 6).

Overall, these experiments are well thought through and they meet the aims of HESS. The research advances hydrological modelling by:

- benchmarking data-driven models (LSTMs) on an hourly timescale

- developing novel model architectures that show state-of-the-art performance

- demonstrate a next step for LSTM-based models to be used in operational forecasting settings

- demonstrate the flexibility of manipulating the loss function in data-driven models to meet different requirements (e.g. timescale consistency).

Furthermore, the availability of the code via the neuralhydrology repository, with an accompanying notebook makes it possible to view the authors assumptions and reproduce the figures in the paper.

**2 SPECIFIC COMMENTS**

I was grateful for the following:

- Figure 2 (P7) is extremely helpful and very professionally made. This is extremely helpful when trying to parse the novel model architecture (MTS-LSTM) proposed by the authors.

- The overt structure outlined on P3 L64-78 is a very helpful signpost to the reader.

- The regularization used to ensure timescale consistency (Sect 2.3.2) is novel and interesting for the target audience of HESS, hydrologists and earth scientists. It confirms the view that the loss function offers huge flexibility to modellers to improve their models for specific use-cases.

- Equation 1 (P8 L180), the annotations to this equation are extremely helpful.

- Table 3 and Table 4 demonstrate an extremely thorough comparison of the models for various metrics and hydrological signatures. This could be used as an example for future benchmarking experiments as an extremely thorough intercomparison, exploring the various facets of the hydrograph.

- Appendix B is a very worthwhile addition, since these negative results can help the field from repeating these results, especially because they turned out to work less well than the model architectures included in the main text. It also outlines the thoroughness of the authors experiments.

- The inclusion of the data and a Jupyter Notebook for readers to reproduce the results is to be applauded. The notebook is well written and the community will be grateful for the time and effort that the authors have put into making their code available and their experiments reproducible. Thank you.

I have a series of comments:

- **P3 L80-87** Are you still using the CAMELS observed discharge or do you now exclusively use the USGS Water Information System REST API values for both hourly and daily evaluation?

- **P3 L81-82** Just to confirm, this is still a "predict timestep including all input data up to time t" rather than a forecast. This is confirmed on P17 L306 but might be worth also including that information here.

- **P4 L101-104** In Section 2.2.1 you describe that you use the NWM v2 Reanalysis product. You describe that this is an hourly product. Do you therefore calculate a daily average of these results to compare against the daily simulations?

- **P6 L131-155** I am still not fully clear on the difference between the sMTS-LSTM and the MTS-LSTM. Can we work to make this slightly clearer in Section 2.3.

- – Do the sMTS-LSTM and the MTS-LSTM receive the same input data?
- – Do both the sMTS-LSTM and MTS-LSTM require two forward passes (L140)
- – It seems that the MTS-LSTM "splits the LSTM into two branches" (L148), which is described as unique to the MTS-LSTM, but then Figure 2 suggests that the sMTS-LSTM also does this splitting but the fully connected layers ($FC_c$, $FC_h$) are simply identity functions.
- – Does the one hot encoding (L141) mean that the LSTM weights are copied in both branches but then zeroed if we are looking at either the hourly or the daily data? If so then why can we not use different input datasets in the sMTS-LSTM as we can in the MTS-LSTM?

There are various solutions. One could: include a table explaining the differences explicitly; include the sMTS-LSTM as its own diagram in Figure 2; or spend more time in Section 2.3 clearly outlining the differences between the two architectures.

- **P6 L154-156** "This architecture makes it clear why we call the other variant "shared" MTS-LSTM: Effectively, the sMTS-LSTM is an ablation of the MTS-LSTM. Both variants have the same architecture, but the weights of the sMTS-LSTM are shared across all per-timescale branches and its state transfer layers are identity operations." I am not what this sentence means. I think it could potentially be clearer for a hydrological audience. My understanding is that an "ablation" means that the sMTS-LSTM is missing something that the MTS-LSTM has, but if they have the same architecture then I am not certain what is missing? From reading and re-reading the difference is something to do with the fully connected layer but I am just a little bit confused about the difference between these two models.

- **P7 L158-169** Related to the misunderstanding of the difference between the sMTS-LSTM and the MTS-LSTM, I am not certain what it means to include multiple datasets and why this could not be done for the sMTS-LTSM. I know that in

the paper: "A note on leveraging synergy in multiple meteorological datasets with deep learning for rainfall-runoff modeling", some of the authors have shown that the LSTM produces more accurate discharge simulations with multiple sources of rainfall information. Is that what is being done in this experiment? Furthermore, if the sMTS-LSTM has the same architecture as the MTS-LSTM (as outlined in the caption to Figure 2), then why can't the sMTS also include new information to the hourly branch? (I am assuming here that the only difference is that the $FC_h$ and $FC_c$ are identity functions rather than linear functions as in the MTS).

- **P7** Related to the comment above. "In the other, we additionally ingested the corresponding day's Daymet and Maurer forcings at each hour. ". Is this data at a daily resolution? If so, does this mean that you are copying the daily inputs 24 times as input for each hour? So if we have hourly NLDAS, You are including Daymet for Day 1 24 times? NLDAS1 + Daymet 1, . . ., NLDAS24 + Daymet1. Apologies if I have misunderstood.

- **P7 L167** "... In the other, we additionally ..." I think it would make sense to explicitly write that you are using the NLDAS forcings AND the Daymet/Maurer forcings. Perhaps something like: "In the other, we ingest the NLDAS forcings as well as the corresponding day's Daymet ..."

- **P13 Table 4**: You write in the Table caption "Bold values highlight results that are not significantly different from the best model in the respective metric or signature ($\alpha$ = 0.001)". I am sure I have misunderstood, but when I look at the Hydrologic Signatures, for example Daily, Q mean. Both the Naive (0.986) and NWM (0.972) results are highlighted. However, Both the sMTS-LSTM (0.985) and the MTS-LSTM (0.984) have values closer to the best model. Is this an artefact of the aggregation? Where the mean is hiding the distribution of Pearson Correlation scores across multiple basins? If so that is fine I just wanted to ensure that this was not a mistake.

- **P14 Table 5** Why do we only see results for the sMTS-LSTM. I believe you have written that it is the "best benchmark model", but is there any other reason to include/exclude the MTS-LSTM? If the experiment was already run it might be an idea to include it, but it is not necessary.

**3 FORMATTING**

I am not certain of the procedure here but I am drawing to your attention in case it is useful.

- **P2 L33**: "... (e.g., Schmidhuber (1991), Mozer (1991))." to "... (e.g., Schmidhuber 1991, Mozer 1991)"

- **P8 L173-174**: "... (e.g., computer vision, Zamir et al. (2020))" to "... (e.g., computer vision, Zamir et al. 2020)"

**4 SUGGESTIONS**

I believe that you are using the terms "look-back window" and "input sequence" interchangeably. Is it perhaps worth using one term consistently through the paper?

- P5 L115 "... look-back windows of 365 days ..."

- P6 L128 "... input sequence of 4320 hours (180 days) ..."

- P6 L137 "... input sequence of $T_D$ time-steps ..."

- P6 L143 "... has access to a large look-back window ..."

- P8 L190 "... achieve a sufficiently long look-back window ..."

---

## Author Comment (AC1) · 8 Feb 2021

**Response to reviewer 2: Thomas Lees**

Comments/Text of reviewer posted in **black**; our answers are posted in **blue**.

**1. OVERVIEW**

This paper seeks to answer the question: "Can a single LSTM model be used to produce accurate and consistent discharge simulations at daily timescales and sub-daily timescales?". The major finding was that yes, you can use a single LSTM to produce daily and hourly predictions. Furthermore, compared with more traditional hydrological models, the MTS-LSTM shows a much smaller performance deterioration when comparing daily simulations (better) to hourly simulations (only slightly worse).

The novel contributions of this paper are threefold:

1. The development of a new multi-timescale LSTM (MTS-LSTM) that produces discharge simulations at both daily and sub-daily timescales (including the flexibility to include arbitrary timescales).

2. The manipulation of the loss function to explicitly account for prior knowledge about the translation between daily and sub-daily timescales. Related to the "hierarchical" nature of these timescales.

3. The benchmarking of a suite of LSTM-based models against the operationally used NOAA National Water Model (NWM).

Research into LSTM based rainfall-runoff modelling has, thus far, mainly focused on simulations at daily timescales. This paper provides a welcome addition to the literature, since sub-daily trends can be important for flood impacts and for water resource managers. The authors focus on producing discharge simulations at a daily timescale and an hourly timescale, although they also show results for 3-hourly and 6-hourly timescales (see Table 2 p9, Table 7 p16).

In order to explore the LSTM architectures that can produce discharge simulations at multiple timescales, the authors suggest three possible avenues (more are included in Appendix B):

- Multiple LSTMs with different timescales, an hourly LSTM and a daily LSTM (naive).

- A "shared" multiple-timescale LSTM (sMTS-LSTM), which overcomes the problems of overly long input sequences, causing long training and inference times for the naive model.

- The MTS-LSTM, which overcomes the problems of the sMTS-LSTM being unable to include different input data for the different timescales.

Both the sMTS-LSTM and the MTS-LSTM are novel contributions to both hydrological modelling, and as far as I am aware, machine learning more generally. My main comment about the paper is that the difference between the sMTS-LSTM and the MTS LSTM could be made clearer.

The authors describe four experiments to demonstrate the usefulness of their newly developed models:

1. Benchmark the MTS models (sMTS-LSTM & MTS-LSTM) against traditional hydrological models (NOAA NWM) and the naive LSTM (which although "naive" is still the most difficult benchmark to compete with). This comparison is thorough and explores accuracy across the hydrograph (see Table 4, Figure 3, Figure 4).

2. Explore the consistency of the MTS models hourly discharge predictions when aggregated to the models daily discharge predictions. The regularisation of the loss function improved the consistency of the sMTS-LSTM.

3. Compare the computational efficiency of the 3 LSTM-based models. The MTS LSTM was the most computationally efficient.

4. Test whether including the same information from different timescales improves model accuracy. The extra information improved forecast accuracy over a range of performance metrics (Table 6).

Overall, these experiments are well thought through and they meet the aims of HESS. The research advances hydrological modelling by:

- benchmarking data-driven models (LSTMs) on an hourly timescale

- developing novel model architectures that show state-of-the-art performance

- demonstrate a next step for LSTM-based models to be used in operational forecasting settings

- demonstrate the flexibility of manipulating the loss function in data-driven models to meet different requirements (e.g. timescale consistency).

Furthermore, the availability of the code via the neuralhydrology repository, with an accompanying notebook makes it possible to view the author's assumptions and reproduce the figures in the paper.

**2. SPECIFIC COMMENTS**

I was grateful for the following:

- Figure 2 (P7) is extremely helpful and very professionally made. This is extremely helpful when trying to parse the novel model architecture (MTS-LSTM) proposed by the authors.

- The overt structure outlined on P3 L64-78 is a very helpful signpost to the reader.

- The regularization used to ensure timescale consistency (Sect 2.3.2) is novel and interesting for the target audience of HESS, hydrologists and earth scientists. It confirms the view that the loss function offers huge flexibility to modellers to improve their models for specific use-cases.

- Equation 1 (P8 L180), the annotations to this equation are extremely helpful.

- Table 3 and Table 4 demonstrate an extremely thorough comparison of the models for various metrics and hydrological signatures. This could be used as an example for future benchmarking experiments as an extremely thorough inter comparison, exploring the various facets of the hydrograph.

- Appendix B is a very worthwhile addition, since these negative results can help the field from repeating these results, especially because they turned out to work less well than the model architectures included in the main text. It also outlines the thoroughness of the authors experiments.

- The inclusion of the data and a Jupyter Notebook for readers to reproduce the results is to be applauded. The notebook is well written and the community will be grateful for the time and effort that the authors have put into making their code available and their experiments reproducible. Thank you.

We would like to thank Thomas Lees for the detailed and thoughtful review. Based on the comments, we have updated our manuscript in several places, most notably to include a more detailed description of the different multi-timescale LSTM variants. We will address each comment individually below (our responses are colored in blue).

**2.1. COMMENTS**

2.1.1. P3 L80-87 Are you still using the CAMELS observed discharge or do you now exclusively use the USGS Water Information System REST API values for both hourly and daily evaluation?

We only use discharge from the REST API in this study. We will add a sentence that clarifies this in the revised manuscript.

2.1.2. P3 L81-82 Just to confirm, this is still a "predict timestep including all input data up to time t" rather than a forecast. This is confirmed on P17 L306 but might be worth also including that information here.

Yes, that is correct: we have a setup for a simulation model. We predict the average daily/hourly discharge at timestep t, using inputs that include the meteorological information of the same timestep t. Since L81-82 are part of the Data section, however, we argue against mentioning this there.

2.1.3. P4 L101-104 In Section 2.2.1 you describe that you use the NWM v2 Reanalysis product. You describe that this is an hourly product. Do you therefore calculate a daily average of these results to compare against the daily simulations?

> Correct. We'll add a sentence to clarify that the lower-resolution predictions are averaged hourly predictions.

2.1.4. P6 L131-155 I am still not fully clear on the difference between the sMTS-LSTM and the MTS-LSTM. Can we work to make this slightly clearer in Section 2.3.

> This relates to Jens Kiesel's comments (questions 2.3.5, 2.3.6 and 2.3.7). We'll briefly answer the specific questions here; for the discussion on a clarified description of sMTS-LSTM we refer to our answer there.

- Do the sMTS-LSTM and the MTS-LSTM receive the same input data?
  > Yes, with the exception that the input of sMTS-LSTM additionally contains a flag that identifies the timescale (one-hot encoding).

- Do both the sMTS-LSTM and MTS-LSTM require two forward passes (L140)?
  > Not quite (though this may to some degree be a matter of interpretation). The sMTS-LSTM does a second forward pass for the timesteps where daily and hourly values overlap. In the MTS-LSTM, the overlapping time steps are handled by different LSTMs, so there is only one forward pass which involves "splitting" the processing across two LSTMs.

- It seems that the MTS-LSTM "splits the LSTM into two branches" (L148), which is described as unique to the MTS-LSTM, but then Figure 2 suggests that the sMTS-LSTM also does this splitting but the fully connected layers ($FC_c$, $FC_h$) are simply identity functions.
  > It is a question of interpretation whether an sMTS-LSTM consists of multiple identical branches or of a single LSTM that is used for multiple input resolutions. We think that the former interpretation nicely highlights the similarity to MTS-LSTM, while the second one might better highlight the differences.

- Does the one hot encoding (L141) mean that the LSTM weights are copied in both branches but then zeroed if we are looking at either the hourly or the daily data? If so then why can we not use different input datasets in the sMTS-LSTM as we can in the MTS-LSTM?

  > The one-hot encoding is related to the inputs (they are essentially additional inputs that are created to differentiate between hourly (e.g., input value 0) or daily (e.g., input value 1). We do not touch the weights at all. We think this also explains the second question. Since the input dimensions do not change (they are the number of meteorological and static inputs + the number of timescales embedded into one-hot encoding), we cannot use different forcings in the sMTS-LSTM model. See also our answer to Jens Kiesel's questions 2.3.6 and 2.3.7..

There are various solutions. One could: include a table explaining the differences explicitly; include the sMTS-LSTM as its own diagram in Figure 2; or spend more time in Section 2.3 clearly outlining the differences between the two architectures.

As said in the answer to Jens Kiesel's questions 2.3.5/2.3.6, we will add a more detailed explanation of the differences between the two variants in the revised manuscript.

2.1.5. P6 L154-156 "This architecture makes it clear why we call the other variant "shared" MTS-LSTM: Effectively, the sMTS-LSTM is an ablation of the MTS LSTM. Both variants have the same architecture, but the weights of the sMTS LSTM are shared across all per-timescale branches and its state transfer layers are identity operations." I am not what this sentence means. I think it could potentially be clearer for a hydrological audience. My understanding is that an "ablation" means that the sMTS-LSTM is missing something that the MTS-LSTM has, but if they have the same architecture then I am not certain what is missing? From reading and re-reading the difference is something to do with the fully connected layer but I am just a little bit confused about the difference between these two models.

The thing that the MTS-LSTM has and the sMTS-LSTM does not is the flexibility to use a different LSTM in the hourly vs. the daily branch. The fully-connected layer is not that important in this context---it is only necessary to make the dimensions of the daily and the hourly states match.

Looking at it conversely, it is maybe clearer that MTS-LSTM is a generalization of sMTS-LSTM: Consider an MTS-LSTM that uses the same hidden size in all branches. This model could learn to use identity matrices as fully-connected layers and equal weights for all LSTM branches, which would make it an sMTS-LSTM (save for the one-hot encoding).

We will rephrase our explanations in the manuscript, together with the improved explanation of MTS-LSTM vs. sMTS-LSTM, to make this more clear.

2.1.6. P7 L158-169 Related to the misunderstanding of the difference between the sMTS-LSTM and the MTS-LSTM, I am not certain what it means to include multiple datasets and why this could not be done for the sMTS-LTSM. I know that in the paper: "A note on leveraging synergy in multiple meteorological datasets with deep learning for rainfall-runoff modeling", some of the authors have shown that the LSTM produces more accurate discharge simulations with multiple sources of rainfall information. Is that what is being done in this experiment?

Partly, yes. There are two possibilities:
(1) using multiple data products that all have the same temporal resolution (this is what's being done in the paper you refer to),
(2) using multiple data products with different temporal resolutions.
Both options can be used with MTS-LSTM. With sMTS-LSTM, the data used at the different time scales must have the same dimensionality (because they're processed by the same LSTM, see question 2.1.4), so option (2) (per-timescale products that may have different amounts of variables) does not work. Option (1) is possible with both architectures.

We will clarify this in the revised manuscript.

Furthermore, if the sMTS-LSTM has the same architecture as the MTS-LSTM (as outlined in the caption to Figure 2), then why can't the sMTS also include new information to the hourly branch? (I am assuming here that the only difference is that the $FC_h$ and $FC_c$ are identity functions rather than linear functions as in the MTS).

The brief explanation is that the input dimensionality would in general not match (see also answer above). Since the weights of the LSTM across all timescales are the same, the number of inputs has to be the same as well (and actually not only the number of inputs, but it should be the *same* inputs). We will rephrase our explanation in the revised manuscript to clarify this.

2.1.7. P7 Related to the comment above. "In the other, we additionally ingested the corresponding day's Daymet and Maurer forcings at each hour." Is this data at a daily resolution?

The brief explanation that clarifies this. To summarize, for every hour of a particular day, we concatenate the daily forcings of the same day as additional inputs.

If so, does this mean that you are copying the daily inputs 24 times as input for each hour? So if we have hourly NLDAS, You are including Daymet for Day 1 24 times? NLDAS1 + Daymet 1, . . ., NLDAS24 + Daymet1. Apologies if I have misunderstood.

Correct (see also the answer to the previous part of this question). We will add a brief explanation that clarifies this.

2.1.8. P7 L167 "... In the other, we additionally ..." I think it would make sense to explicitly write that you are using the NLDAS forcings AND the Daymet/Maurer forcings. Perhaps something like: "In the other, we ingest the NLDAS forcings as well as the corresponding day's Daymet ..."

We will rephrase the sentence to be more explicit.

2.1.9. P13 Table 4: You write in the Table caption "Bold values highlight results that are not significantly different from the best model in the respective metric or signature (α = 0.001)". I am sure I have misunderstood, but when I look at the Hydrologic Signatures, for example Daily, Q mean. Both the Naive (0.986) and NWM (0.972) results are highlighted. However, Both the sMTS-LSTM (0.985) and the MTS LSTM (0.984) have values closer to the best model. Is this an artefact of the aggregation? Where the mean is hiding the distribution of Pearson Correlation scores across multiple basins? If so that is fine I just wanted to ensure that this was not a mistake.

The highlighting is correct, and it is indeed an artefact of the aggregation. For signatures (not metrics), the table shows the Pearson correlation with observed values. The bold font highlights models for which the results were not significantly different to those of the model with the highest Pearson correlation. The significance test (Wilcoxon) has the null hypothesis that the differences between two populations are symmetric around zero. Since

Pearson correlation doesn't operate on these differences, there exist cases where the test is significant but the Pearson correlation is close to that of the reference group. Had we chosen the Spearman correlation (which is based on ranks), sMTS-LSTM would have had a slightly higher correlation coefficient than Naive.

2.1.10. P14 Table 5 Why do we only see results for the sMTS-LSTM. I believe you have written that it is the "best benchmark model", but is there any other reason to include/exclude the MTS-LSTM? If the experiment was already run it might be an idea to include it, but it is not necessary.

We only reported results for sMTS-LSTM since it was the best model in the above benchmarking. While we have not done it so far, the experiment would certainly be possible for MTS-LSTM, too.

**3. FORMATTING**

I am not certain of the procedure here but I am drawing to your attention in case it is useful.

3.1. P2 L33: "... (e.g., Schmidhuber (1991), Mozer (1991))." to "... (e.g., Schmidhuber 1991, Mozer 1991)"

3.2. P8 L173-174: "... (e.g., computer vision, Zamir et al. (2020))" to "... (e.g., computer vision, Zamir et al. 2020)"

Thank you for pointing this out. We will change the references to the correct format.

**4. SUGGESTIONS**

I believe that you are using the terms "look-back window" and "input sequence" interchangeably. Is it perhaps worth using one term consistently through the paper?

- P5 L115 "... look-back windows of 365 days ..."
- P6 L128 "... input sequence of 4320 hours (180 days) ..."
- P6 L137 "... input sequence of $T_D$ time-steps ..."
- P6 L143 "... has access to a large look-back window ..."
- P8 L190 "... achieve a sufficiently long look-back window ..."

We agree that it makes sense to work on a more consistent use of the two terms. The terms are almost identical, but there are slight differences: A long input sequence does not have to mean that the model looks far into the past (if the resolution is high). Also, look-back seems like a nice way to refer to input sequences regardless of their timescale, whereas we usually associate

input sequences with a fixed timescale (e.g., hourly). We will try to follow this distinction in the revised manuscript.

---

## Author Comment (AC2) · 8 Feb 2021

**Response to reviewer 1: Jens Kiesel**

Comments/Text of reviewer posted in **black**; our answers are posted in **blue**.

**1. GENERAL COMMENTS**

The manuscript "Rainfall–Runoff Prediction at Multiple Timescales with a Single Long Short-Term Memory Network" (LSTM) by Martin Gauch et al. presents an extension of LSTM hydrological models to sub-daily time steps. In previous publications, LSTMs as hydrological models were used on a daily time step. The authors explore multiple approaches to achieve a 'multi-timescale' model, of which three (naive LSTM, sMTS LSTM, MTS-LSTM) are evaluated in more detail and less promising experiments are briefly explained in an Annexe. Similar to previous applications of LSTMs, the models are applied at the CAMELS dataset, encompassing 516 basins across the contiguous USA, where hourly data is available. Results are compared to the NOAA National Water Model (NWM) and show that all LSTMs architectures outperform the NWM. The authors suggest that the MTS-LSTM provides most flexibility for future use.

The manuscript is generally well written and structured, figures and tables support the results. Having a more process-based hydrological background, I nevertheless read the paper with interest and believe it fits well in the scope of HESS. I see the work as highly relevant, especially in the field of flood modelling and (eventually) forecasting, but also generally in the application of LSTMs at different temporal resolutions. However, especially regarding the latter, I think the authors should invest more work to improve the usefulness of the paper. Please find below more detailed comments, questions and suggestions that hopefully initiate a fruitful discussion and help in improving the paper.

We would like to thank Jens Kiesel for his detailed and thoughtful review. Based on his comments, we have prepared a revised manuscript that (among other changes) provides a more accessible review of related work in machine learning and gives a more detailed description of the differences between our proposed multi-timescale LSTM variants. In the following, we address each comment individually. Our answers are colored in blue.

**2. SPECIFIC COMMENTS**

**2.1. ABSTRACT**

I suggest to mention the difficulties and challenges applying the models (parameter estimation) and discuss the work still to be done regarding different time scales (e.g. generalization of parameters)

**2.2. INTRODUCTION**

I think you are missing a research gap in your introduction which is important to apply the LSTM for different time steps, since there seems to be a time step dependency of model parameters / hyperparameters (e.g. hidden size, sequence length, batch size, forget gate bias, learning rate?, others?). Due to the computationally expensive training of LSTMs, knowing which ones need to be adjusted, in about which range and identifying ideal values is essential. I would like to see this topic included in the "contributions" you list at the end of the introduction (and therefore also more prominently in the respective chapters).

2.2.1. p.2 l.29-41: I think this section is difficult to understand for a reader without firm neural networks background. Particularly phrases like: "partitions a recurrent neural network into layers with individual clock speeds", "process irregularly sampled inputs by means of a time gate that only attends to the input at steps of a learned frequency", "the approach depends on a binary decision that is only differentiable through a workaround". I acknowledge that your paper cannot serve as an introduction to the topic. I have no clear suggestion other than making this paragraph more accessible to readers with a hydrological background through using less specialized jargon, if possible.

Thank you for the feedback. We agree with this analysis of the language. In the revised version, we will explain the related work and its connections to MTS-LSTM in terms that are more familiar to the intended audience as follows (formerly L29-41):

"More recently, Koutnik et al. (2014) proposed an architecture that efficiently learns long- and short-term relationships in time series. They partition the internals of their neural network into different groups, where each group is updated on individual, pre-defined, intervals. However, the whole sequence is still processed on the highest frequency, which makes training slow. Chung et al. (2016) extended the idea of hierarchically processing the different timescales: In their setup, the model can adjust its updating rates to the current input, e.g., to align with words or handwriting strokes. Unfortunately, the binary decision whether to make an update complicates the procedure, since it can only be trained through a workaround. Neil at al. (2016) proposed an LSTM architecture with a gating mechanism that allows state and output updates only during time slots of learned frequencies. For time steps where the gate is closed, the old state and output are reused. This helps discriminate superimposed input signals, but is likely unsuited for rainfall--runoff prediction because no

aggregation takes place while the time gate is closed. In a different research thread, Graves et al. (2007) proposed a multidimensional variant of LSTMs for problems with more than one temporal or spatial dimension. Framed in the context of multi-timescale prediction, one could define each timescale as one temporal dimension and process the inputs at all timescales simultaneously. Like the hierarchical approaches, however, this paradigm would process the full time series at all timescales and thus lead to slow training."

2.2.2. p.2 l.45 and 47: You write that Araya et al. predicted wind speed at "multiple timescales". Then you mention that your objective is "multiple outputs, one for each target timescale". I don't understand the difference between that.

Given our current formulation it is understandable that the difference was unclear. By "they predicted wind speed given input data at multiple timescales," we meant that the *input* data to their model is at multiple timescales (e.g., hourly and daily input values). The *output*, however, is only at the hourly timescale. MTS-LSTM, in contrast, produces a distinct output for each timescale (e.g., discharge prediction at hourly and daily timescale). We will rephrase the sentence to clarify this.

2.2.3. p.2 l.54ff: I see the capability to process input data in irregular intervals as an advantage. Think of satellite products that have different data gap length (e.g. soil moisture or altimetry products combining multiple sensors). You can discuss this further, but at least I suggest to write on p.3 l.77: "...LSTM can ingest individual and multiple sets of forcings each having regular time intervals for each target timescale. This closely resembles..."

Thank you, we will adopt the suggestion.

2.2.4. p.3 l.70-72, 74-75: I suggest not to mention the results of your study in the introduction

We understand your suggestion. However, we believe that a brief description of the overall results in the introduction helps readers navigate the manuscript and emphasizes why the contributions are meaningful. Also, it serves people who only read the introduction/contributions and conclusion for a high-level overview on the paper. That said, if our view is strongly opposed by the editor or reviewers, we are open to revise the introduction accordingly.

2.2.5. p.3 l.64-78: These three paragraphs reveal that your introduction could be structured a bit better, ideally introducing the reader to these three problems/research gaps that need to be solved for "Rainfall–Runoff Prediction at Multiple Timescales with a Single Long Short-Term Memory Network". You have motivated the first paragraph, but the second and third 'contribution' that you list appears a bit unexpected since your previous introduction does not resemble that structure. For instance, instead of referring to sections later in the paper, I believe it would be better to introduce the reader to the problem of inconsistencies. You briefly mention this on p.2 l.27-28 for conventional hydrological models, but this can be extended, especially targeted on machine learning.

Thank you for these suggestions. In the revised manuscript, we will introduce the problems of inconsistency and of per-timescale data products in the introduction, such that they appear less unexpectedly in the contributions.

**2.3. DATA AND METHODS**

2.3.1. p.4 l.92-94: The distinction into training, validation and test is not fully clear to me. You use the validation period to evaluate different architectures and to select model hyperparameters. Could you elaborate on the reason why the evaluation of architecture and the hyperparameter selection cannot/should not be done during the training period?

Using a three-way data split is standard practice in machine learning. This is so that hyperparameters are not chosen based on the test set, which would be cheating.
The purposes of the different periods are as follows:
- The **training period** is used to fit the model, given a set of hyperparameters.
- This model is then applied to the **validation period** to evaluate its accuracy on previously unseen data. The training and validation periods can be used to adjust the model. To find the "best" hyperparameters, this process is repeated a number of times (with different hyperparameters) and the model that achieves the best validation accuracy is selected.
- Only once a final model is selected, we apply it to the **test period**. This way, the model has never seen the test data before and we can be sure that we didn't overfit our model to the benchmarked metric.

If we had only a training and a test period and selected the hyperparameters on the training period, our model would likely not generalize to unseen data (such as the test period), because we only ever focused on modeling the training period. In the extreme, we would train our model to memorize every data point of the training period and get a perfect fit but terrible accuracy on any other data.

We have extended the paragraph in the revised manuscript to explain the purposes of the training, validation, and test periods in more detail.

2.3.2. p.4 l.101ff: Can you describe the datasets used in NWM and the basic characteristics (e.g. spatial application range, calibration strategy and performance) of the v2 reanalysis product?

Unfortunately, this information is not fully public. We will add the following paragraph to the description of NWM: "For these predictions, NWM was calibrated on meteorological NLDAS forcings for around 1500 basins and then regionalized to around 2.7 million outflow points (source: personal communication with NOAA scientists)."

2.3.3. p.5 Fig1: please also mention what "x" and "+" represent

They represent element-wise multiplication ("x") and addition ("+"). We will clarify this in the revised manuscript.

2.3.4. p.6 l.127-130: I am particularly interested in how you tuned these parameters and how you decided which parameters to adjust and which ones not. As you mention, the LSTM application is computationally expensive and parameter selection and ranges are therefore important. Therefore, I would rather want to see Appendix D in the main text, and include information why certain parameters are time step dependent and others not.

The decision which parameters to adjust and within which bounds to adjust them is unfortunately largely based on personal experience with LSTMs. To our knowledge, there exist no rules that are universally applicable and agreed upon for the hyperparameter tuning with these models (beyond basic principles like train-validation-test splits, see 2.3.1).

As for timestep-dependent vs. -independent parameters: Are you referring to hyperparameters that are (or can be) different for each *timescale*? In theory, with MTS-LSTM, one could use different parameters for each LSTM branch (e.g., hourly and daily branch) with respect to almost all hyperparameters. The question therefore is for which hyperparameters it may make sense to use different values in the different branches. E.g., for hidden size, we do not see a reason to choose vastly different sizes for the different timescales, since each LSTM branch models a similar process. The only hyperparameter that we *did* choose per timescale is the sequence length, because it defines the point in time where the daily branch hands off to the hourly branch.
For sMTS-LSTM, there are some additional restrictions: Since the daily and hourly LSTM branches use the same weights, they cannot have different hidden sizes (because that would entail different amounts of weights).

Given these considerations, unless there is strong opposition from the editor or reviewers, we would prefer to keep Appendix D in the appendix. Since the concrete hyperparameter choices are very particular to our evaluated models and setup, we think keeping them in the appendix helps to avoid the impression that these may be universally applicable choices for LSTMs (even a dataset of different size may lead to other parameters being better suited). A detailed description of the possibilities, pitfalls, and empirical experience of hyperparameter tuning would be material for a publication in itself.

Also, In Table D1, it seems you ended up with 336 hrs sequence length for both architectures. Would an even longer sequence length lead to better results? What is the tradeoff between higher sequence lengths and computational costs?

Longer sequences do not necessarily lead to better results. The longer the hourly input sequence, the longer the overall input time series will be (because the transition from daily to hourly inputs will happen earlier). Such longer time series are harder for LSTMs to

process, because they need to learn dependencies across many time steps. Further, the additional time steps will increase the computational demand of the model. In the extreme, if we'd use an hourly sequence length of 365*24, the hourly branch in the MTS-LSTM would require as much computation as the naive hourly LSTM.

Conversely, shorter hourly input sequence lengths reduce computational complexity (because more of the input is processed at the daily resolution, leading to shorter time series). A too short hourly input sequence, however, will remove information from the inputs that is necessary to generate high-resolution hourly predictions. That said, in our experiments, we did not observe high sensitivity of the model accuracy with regards to the exact hourly input sequence length.

2.3.5.    p.6 l. 146-156: Could you explain why these two different LSTM architectures were developed? What are the expected advantages/disadvantages?

sMTS-LSTM is a special case of MTS-LSTM where the different LSTM branches are actually the same LSTM (they have the same structure and weights, see answer to 2.3.6), and therefore they model the same input--output relationships. A-priori, it seems reasonable that some relationships that govern daily prediction hold, to some extent, for hourly predictions as well. Hence, it may be easier to learn these dynamics in a single LSTM branch that processes the different timescales (as done in sMTS-LSTM) than to learn them multiple times, once in each branch (as done in MTS-LSTM). On the other hand, however, there are also differences in how daily vs. hourly data are processed, and these may be easier to learn in a branch that focuses on one timescale (MTS-LSTM) than in a branch that's shared across timescales (sMTS-LSTM).

We agree that the current description does not motivate the two MTS-LSTM variants, and we will include this reasoning in the revised manuscript (formerly L136-156). The following revised text also clarifies the differences between MTS-LSTM and sMTS-LSTM (see questions 2.3.6, 2.3.7):

"The first model, shared multi-timescale LSTM (sMTS-LSTM), is a simple extension of the naive approach. Intuitively, it seems reasonable that the relationships that govern daily predictions hold, to some extent, for hourly predictions as well. Hence, it may be possible to learn these dynamics in a single LSTM that processes the time series twice: Once at a daily resolution and again at an hourly resolution. Since we model a damped system, where the resolution of long-past time steps is less important, we can simplify the second (hourly) pass by reusing the first part of the daily time series. This way, we only need to use hourly inputs for the more recent time steps, which yields shorter time series that are easier to process. From a more technical point of view, we first generate a daily prediction as usual---the LSTM ingests an input sequence of $T_D$ time steps at daily resolution and outputs a prediction at the last time step (i.e., sequence-to-one prediction). Next, we reset the hidden and cell states to their values from time step $T_D-T_H/24$ and ingest the hourly

input sequence of length T_H to generate 24 hourly predictions that correspond to the last daily prediction. In other words, we reuse the initial daily time steps and use hourly inputs only for the remaining time steps.

In summary, we perform two forward passes through the same LSTM at each prediction step: one that generates a daily prediction and one that generates 24 corresponding hourly predictions. Since the same LSTM processes input data at multiple timescales, it needs a way to identify the current input timescale and distinguish daily from hourly inputs. For this, we add a one-hot timescale encoding to the input sequence. The key insight with this model is that the hourly forward pass starts with LSTM states from the daily forward pass, which act as a summary of long-term information up to that point. In effect, the LSTM has access to a large look-back window but, unlike the naive hourly LSTM, it does not suffer from the performance impact of extremely long input sequences.

The second architecture, illustrated in Fig. 2, is a more general variant of the sMTS-LSTM that is specifically built for multi-timescale predictions, hence, we call it the *multi-timescale LSTM* (MTS-LSTM). Its architecture stems from the idea that the daily and hourly predictions may behave so differently that it is challenging for one LSTM to learn both dynamics, as the sMTS-LSTM would have to. Instead, it may be easier to process the inputs in individual LSTMs per timescale. To reuse daily processing steps towards hourly predictions, MTS-LSTM does not perform two forward passes (as sMTS-LSTM does).Instead, it splits an individual hourly LSTM branch off of the daily LSTM after the initial daily time steps (see Fig. 2). Expressed more technically: we first generate a prediction with an LSTM acting at the coarsest timescale (here: daily) using a full input sequence of length T_D (e.g., 365 days). Next, we reuse the daily hidden and cell states from step T_D-T_H/24 as the initial states for an LSTM at a finer timescale (here: hourly), which generates the corresponding 24 hourly predictions. Since the two LSTM branches may have different hidden sizes, we feed the states through a linear state transfer layer (FC_h, FC_c) before reusing them as initial hourly states. In this setup, each LSTM branch only receives inputs of its respective timescale, hence, we do not need to one-hot encode the timescale. This architecture makes it clear why we call the other variant "shared" MTS-LSTM.

Effectively, the sMTS-LSTM is an ablation of the MTS-LSTM: One can see the sMTS-LSTM as an MTS-LSTM where the different LSTM branches all share the same set of weights and the states are transferred without any additional computation (i.e., the transfer layers are identity functions). Conversely, the MTS-LSTM is a generalization of sMTS-LSTM: Consider an MTS-LSTM that uses the same hidden size in all branches. In theory, this model could learn to use identity matrices as transfer layers and to use equal weights in all LSTM branches. Save for the one-hot encoding, this would make it an sMTS-LSTM."

The last sentence is crucial for the understanding of the differences, I believe "weights of the sMTS-LSTM are shared across all per-timescale branches and its state transfer layers

are identity operations." What is an identity operation?

An identity operation is a function that outputs the input value(s). We will rephrase this to "states are transferred without any additional computation (i.e., the transfer layers are identity functions)" which we hope to be clearer. For more on the difference between MTS-LSTM and sMTS-LSTM, see our answers to questions 2.3.6/2.3.7.

2.3.6.    p7. Figure 2: I understood from the text that both the sMTS-LSTM and MTS-LSTM are branching out at each day into hourly predictions. The MTS-LSTM predicts 24 hours, using 72hrs sequence length. Is this the same for the sMTS-LSTM?

It is correct that both MTS-LSTM and sMTS-LSTM predict 24 hours using 72 hours of hourly input sequence length. The first part of the statement ("both the sMTS-LSTM and MTS-LSTM are branching out at each day into hourly predictions") could be misunderstood: For the prediction of any given day, the hourly LSTM branches off of the daily LSTM only *once* (72h before the last time step). But, if we predict subsequent days to obtain a time series of predictions, the branching point will shift by one day as the predicted day moves forward. All of this holds for both MTS-LSTM and sMTS-LSTM .

The difference between sMTS-LSTM and MTS-LSTM is difficult to understand from just the figure caption. I think it would help to construct the illustration for both architectures to visualize the differences, if possible including the different weights for the MTS-LSTM and the similar weights for the sMTS-LSTM in the diagram.

Unfortunately, we could not find a good way to illustrate the difference between MTS-LSTM and sMTS-LSTM. The basic idea of shared weights is that the daily LSTM branch will behave identical to the hourly branch (if applied to the same inputs). Since the LSTM blocks have a complex internal structure (depicted in Figure 1), it is hard to explicitly show the model weights.

Maybe an alternative perspective on sMTS-LSTM can clarify the setup:

Another way to think of sMTS-LSTM is a single LSTM without any branches. The model works as follows:
1) First, we add timescale flags to the input data:
- We concatenate each timestep of the daily inputs with a one-hot encoding of "daily timescale" (e.g., a vector $(1, 0)^T$).
- We concatenate each timestep of the hourly inputs with a one-hot encoding of "hourly timescale" (e.g., a vector $(0, 1)^T$).
2) Then, we ingest the full daily input sequence into the LSTM. This gives us a daily prediction.
3) Next, we re-initialize the LSTM with the hidden and cell state from 3 days ago.

4) Finally, we ingest the last 72 steps of hourly data into the LSTM. This gives us 24 hourly predictions.
Described from this angle, the differences to MTS-LSTM are:
- MTS-LSTM does not need step (1), since there is no need for timescale flags.
- With MTS-LSTM, steps (3) and (4) operate on a *different* LSTM than step (2).

Our rephrased descriptions of MTS-LSTM and sMTS-LSTM (see 2.3.5) should be clearer on these differences.

2.3.7.  p.7 l.158: I don't understand why the MTS-LSTM is more flexible in terms of input data than the sMTS-LSTM. In the sMTS-LSTM section you write (p.6 l.139): "we....ingest the hourly input sequence of length TH to generate 24 hourly predictions that correspond to the last daily prediction." Looking at Fig 2, to me this is similar in the MTS-LSTM, where the daily forcings have an effect until the hourly branch starts and then no update using the daily forcings/predictions seems to be made in the hourly branch. Therefore, effectively, you use the daily data until the model branches out and then you use the hourly forcings only? Again, I think it would help to show both architectures in Fig 2.

Our explanation to questions 2.3.5 and 2.3.6 probably clarify this. The reason why sMTS-LSTM cannot use different data products (or even different amounts of data products) for different timescales is that steps (2) and (4) use the *same* LSTM. A single LSTM has a fixed input dimensionality and is therefore unable to process input vectors of varying size. Different meteorological data products may have different numbers of variables, so they cannot be processed by the same LSTM. Since missing explanations from our side seem to inhibit understanding of this point, we include a better description of the differences as outlined in our answers to 2.3.5/2.3.6.

2.3.8.  p.8 l.170-184: If I understand it correctly, adding the term into the loss function 'encourages' the model to minimize the difference between daily and sub-daily simulation. But similar to the NSE, this ideal value may not be reached, ending up with a model that is not consistent - even if you put an exceptionally high weight on the mean squared difference?

Yes, this is exactly how the loss works (though the higher you weigh the difference, the more likely will the model learn to generate consistent predictions---but this will come at the cost of poor predictions: e.g., predicting constant 0 is consistent but not useful).

Is there a reason why you don't 'force' consistency across timescales? E.g. when looking at Figure 2 I imagine you could add a function (e.g. simple multiplication of a term) that scales either the daily or the sub-daily prediction (or the average between the two) so that both match the consistency criteria (I now notice that may be similar to what you did in "B1 Delta Prediction")?

Enforcing consistency is in principle possible. And as you note, we tried a number of

approaches to achieve this in Appendix B. However, in our experiments, these approaches yielded worse predictions than MTS-LSTM, so we did not further pursue them.

2.3.9. p.9 Table 2: it is a bit confusing to have these different sequence lengths. In the previous section it is 72hrs, here 168hrs, in Table D1 it is 336hrs. Can you harmonize this or explain why there are these differences?

We will clarify this in the revised manuscript.The reasons for the differences are the following:

The 72h in Figure 2 are just for the purpose of illustration: In order to keep the figure tidy, we did not want to show too many "LSTM boxes" after the split into daily and hourly LSTM, so we decided on three days, which translates to 72 hours.

The 336h in Table D1 is what we use in most experiments. This is the outcome of our hyperparameter search for daily--hourly prediction.

The 168h in Table 2 are part of a model that demonstrates how MTS-LSTM can be used beyond daily--hourly modeling and also predict other timescales. The concrete input sequence lengths in Table 2 are somewhat arbitrary (and not hyperparameter-tuned), since our point in this section is less to achieve the best possible NSE but rather to show the flexibility of MTS-LSTM.

We will clarify this by adding "we chose this value for the sake of a tidy illustration; the benchmarked model uses $T^H=336$" to the caption of Fig. 2 and "The specific input sequence lengths are chosen somewhat arbitrarily, as our intention is to demonstrate the general capability rather than to achieve the best possible NSE." to the caption of Table 2.

**2.4. RESULTS**

2.4.1. p.9 l.210: that means running ten seeds based on the parameterization in bold in Table D1? If so, I'd add this here.
Correct. We will add this in the revised manuscript.

2.4.2. p.9 l.219: I find this particularly interesting when thinking about hydrological processes. The model parameter values (hidden and cell states) of the last coarse time step (Td - Th/24) are basically your boundary condition/initial state for the hourly model. It seems a bit counterintuitive that the sMTS-LSTM performs better than the naively trained full hourly LSTM. So the 'error' you introduce through the daily average initial state must be insignificant (due to a sufficiently long sequence length?).

Yes. This is a nice way to think about the modelling system and relates to our motivation of the MTS-LSTM architecture. The idea is that in a damped system, early time steps do not need to be processed at the high target resolution.

Particularly in small basins and for flood peak prediction, this may not always be the case. A plot showing the spatial differences in performance between the naively trained LSTM, the sMTS-LSTM and MTS-LSTM (e.g. similar to Fig 4) could reveal if/where these differences exist. I'd however not be surprised if this plot will show no pattern due to input data uncertainty and randomness in the LSTM and the small performance difference between the LSTM types.

Figure 1 below shows the spatial patterns of the difference between the NSE of sMTS-LSTM and Naive (hourly) predictions together with basin size (indicated by the marker size). As predicted by the reviewer, we cannot see any outstanding patterns that would indicate relationships between basin size and NSE difference.

[Figure]

Figure 1: NSE differences between sMTS-LSTM and Naive (hourly). Marker size encodes basin size.

2.4.3.  p.14 l.237-250: Interestingly, the Naive LSTM deviates most - probably because the sMTS-LSTM and the MTS-LSTM use recent states from the daily model and are therefore 'closer' to the daily model's flow (volume) prediction?

Yes, this is a plausible assumption, since the shared states will likely make it easier for the model to minimize the consistency term (in some sense one might see the shared states as an inductive bias towards consistency). Unfortunately, we do not see a way to prove it.

The beneficial influence on the NSE could arise because you are introducing a 'physically plausible' constraint in the model which 'helps' adapting the network to the processes? (see

also my comment to p8. l.170-184). That is an interesting prospect and if true, could mean adding more of such physical constraints (e.g. global water balance closure) could improve the LSTM even further?

We agree with this line of reasoning and will include it in the revised manuscript. The physically plausible architecture might benefit the predictions. In machine learning, this is sometimes referred to as "inductive bias". There is indeed ongoing work in hydrology to add such constraints in the form of physical constraints or adaptations into ML/LSTM models (for water balance, we'd like to refer to [1]). However, to our knowledge, so far no modification has improved the overall model performance w.r.t. the NSE.

**2.5. CONCLUSIONS**

2.5.1.   p.16 l.292: it depends on how the NWM was calibrated and what the main purpose is (see also comment to p.4 l.101ff)

We refer to our answer to question 2.3.2 on details of the calibration procedure. Unlike the LSTM-based models, NWM was calibrated only for hourly and not for daily predictions, which may affect the accuracy. This, however, only corroborates our point: Had NWM been explicitly calibrated for daily predictions, we'd expect *equal or better* daily NSEs---and therefore the gap between daily and hourly quality would only grow. Further, since the model was calibrated w.r.t. NSE (like the LSTM-based models) by people who are experts in its usage (NOAA scientists), we would argue that our performance comparison is valid.

2.5.2.   p.17 l.293: I understand and agree. But given that LSTMs perform so well for hydrological modelling, efforts should be made to generalize the hyperparameter values for different time steps. I believe you were not sufficiently confident with your tests to deduce general rules for the hyperparameter settings (and that may be a reason why this analysis ended up in the Annexe). But I think it would help the future application of LSTMs if you could give a summary of your experience: e.g. which parameters are time-step dependent, should a parameter increase or decrease with increasing/decreasing time steps, what if someone applies an even coarser time step (monthly)?

First, we would like to reiterate our comment from 2.3.4: We are unaware of any non-trivial universal rules on hyperparameter selection, and would therefore like to avoid the impression that our final parameters are ideal for other tasks and datasets. That said, researchers who work on a similar task and dataset could certainly take our parameters as a starting point for their own tuning procedure.
More specifically to the point of dependence on timescales, only hidden size and sequence length seem meaningful to choose per timescale (though one could maybe come up with scenarios where different learning rates make sense, but we didn't explore this). As stated in the answer to question 2.3.4, we did not see strong sensitivity with regards to the choice of sequence length (but, again, to achieve the best possible NSE, one will have to

hyperparameter-tune the model to the specifics of the application). For hidden size, we do not see a reason to choose vastly different sizes for the different timescales, since each LSTM branch models a similar process.

2.5.3. p.17 l.296-298: I know the differences are not statistically significant, but can you speculate on why the models are ranked in that order? Somehow the naive hourly LSTM seems not to be able to use this additional information content, or the half year sequence length is not sufficient to depict all states (e.g. groundwater storages may need longer sequence length in some catchments)?

Unfortunately, we cannot make any conclusive statements. One plausible explanation could be that the long input sequences make it harder for the naive LSTM to learn all relationships that exist in the data, while sMTS-LSTM only needs to derive relationships from shorter time series (and the lower resolution doesn't matter much, since it's only low for time steps far in the past). A theoretical explanation could be given by the vanishing gradient phenomen, which is the reason why it is hard for LSTMs to learn dependencies over very long sequences (e.g., more than 1000 time steps).

2.5.4. p.17 l.299-305: Can you speculate why the daily forcings to the hourly MTS-LSTM improve the performance?

We believe that this has essentially the same reason why multiple daily forcings improve daily predictions (already known from previous publications, [2]): Each data product has its individual errors, and given multiple products, the LSTM can intelligently combine the information to counteract these errors.The fact that in this case we use daily data for hourly predictions might reduce this impact, but clearly it does not fully remove it. This may be supported by some degree of smoothness across time: If the daily product says there is low temperature, most (if not all) hours will have had low temperatures, too.

2.5.5. I believe there is more research to be done that you can mention here? E.g. a thorough investigation of time step-dependency of hyperparameters, find measures to use physical constraints in the LSTM (e.g. the regularization)

We agree that there is more research to be done, and we'll add a sentence on physical constraints in the revised paper. Other areas of future research include exploring the potential of uncertainty estimation at multiple frequencies (as opposed to point estimates) and the exploration of architectures that pass information not just from coarse to fine timescales, but also vice versa (similar to our preliminary ResNet experiments that we report in the appendix).

**3. TECHNICAL CORRECTIONS**

3.1.  once introduced, you can stick to the abbreviations (e.g. NWM, MTS-LSTM)

We'll change the revised manuscript to consistently use the abbreviations after their introduction.

3.2.  p.3 l.58-60: I think you can refer to Appendix C here

We'll add the reference in the revised submission.

3.3.  p.5 l.118: ...half a year...

Yes, we'll change this.

3.4.  p.8 l.191-192: it is uncommon to mention results in the methods

Agreed, we'll remove the sentence on results.

3.5.  p.8 l.199: this link is supplied here for the third time. Not sure if this is how HESS wants to have references to URLs.

Agreed, the repeated footnote is not necessary. We'll remove it and change the footnotes to citations as per HESS standards.

3.6.  p.9 l.215: 'even the naive ones' - the naive LSTM acts as a benchmark, so it is expected it performs better than (s)MTS?

Yes and no: Yes, it is a benchmark (in the sense of being the most straight-forward way of achieving hourly predictions with LSTMs). But, as explained in the Methods section, we expected the hourly naive LSTM to be problematic since it has to work better or worse than MTS-LSTM.

3.7.  p.9 l.216: I think it is fair to add that this worse performance on hourly is much more visible at the NWM

We'll add this in the revised manuscript.

3.8.  p.17 l.311-312: I find this first sentence difficult to understand. If possible, split in two

We'll split the sentence and slightly rephrase to clarify.

**4. REFERENCES**

[1] Hoedt, P.-J., Kratzert, F., Klotz, D., Halmich, C., Holzleitner, M., Nearing, G., Hochreiter, S., Klambauer, G.: MC-LSTM: Mass-conserving LSTM, https://arxiv.org/abs/2101.05186, arXiv, 2021.

[2] Kratzert, F., Klotz, D., Hochreiter, S., and Nearing, G. S.: A note on leveraging synergy in multiple meteorological datasets with deep learning for rainfall-runoff modeling, Hydrol. Earth Syst. Sci. Discuss. [preprint], https://doi.org/10.5194/hess-2020-221, in review, 2020.

---

## Author Response (AR2)

**Authors' Response**

Again, we would like to thank the reviewer and the editor for their feedback.

Regarding Jens Kiesel's remaining comments: We agree that there is potential for future research around general guidelines for hyperparameters. As suggested, we have removed the sentence "Again, this is standard practice for machine learning" from the manuscript.